# Remove Symmetries to Control Model Expressivity and Improve Optimization

**Liu Ziyin**[1,2,*]**, Yizhou Xu**[3,*]**, Isaac Chuang**[1,4]

[1]*Research Laboratory of Electronics, Massachusetts Institute of Technology*
[2]*Physics & Informatics Laboratories, NTT Research*
[3]*Computer and Communication Sciences, École Polytechnique Fédérale de Lausanne*
[4]*Department of Physics, Massachusetts Institute of Technology*

## Abstract

When symmetry is present in the loss function, the model is likely to be trapped in a low-capacity state that is sometimes known as a "collapse." Being trapped in these low-capacity states can be a major obstacle to training across many scenarios where deep learning technology is applied. We first prove two concrete mechanisms through which symmetries lead to reduced capacities and ignored features during training and inference. We then propose a simple and theoretically justified algorithm, *syre*, to remove almost all symmetry-induced low-capacity states in neural networks. When this type of entrapment is especially a concern, removing symmetries with the proposed method is shown to correlate well with improved optimization or performance. A remarkable merit of the proposed method is that it is model-agnostic and does not require any knowledge of the symmetry.

## 1 Introduction

The unprecedented scale and complexity of modern neural networks, which contain a vast number of neurons and connections, inherently introduce a high degree of redundancy in model parameters. This complexity in the architecture and the design of loss functions often implies that the loss functions are invariant to various hidden and nonlinear transformations of the model parameters. These invariant transformations, or "symmetries," in the loss function have been extensively documented in the literature, with common examples including permutation, rescaling, scale, and rotation symmetries (Simsek et al., 2021; Entezari et al., 2021; Dinh et al., 2017; Neyshabur et al., 2014; Tibshirani, 2021; Zhao et al., 2023; Godfrey et al., 2022; Ziyin et al., 2024).

A unifying perspective to understand how these symmetries affect learning is the framework of reflection symmetries (Ziyin, 2024).[1] A per-sample loss function $\ell$ possesses a $P$-reflection symmetry if for all $\theta$, and all data points $x$,

$$\ell((I - 2P)\theta, x) = \ell(\theta, x). \tag{1}$$

The solutions where $\theta = (I - P)\theta$ are the symmetric solutions with respect to $P$. It has been shown that almost all models contain quite a few reflection symmetries. Permutation, rescaling, scale, and rotation symmetries all imply the existence of one or multiple reflection symmetries in the loss.

Recent literature has shown that symmetries in the loss function of neural networks often lead to the formation of low-capacity saddle points within the loss landscape (Fukumizu, 1996; Li et al., 2019). These saddle points are located at the symmetric solutions and often possess a lower capacity than the minimizers of the loss. When a model encounters these saddle points during training, the model parameters are not only slow to escape them but also attracted to these solutions because these the gradient noise also vanish close to these saddles (Chen et al., 2023). Essentially, the model's learning process stagnates, and it fails to achieve optimal performance due to reduced capacity. However, while many works have characterized the dynamical properties of training algorithms close to symmetric solutions, no methods are known to enable full escape from them.

Because these low-capacity saddles are created by symmetries, we propose a method to explicitly remove these symmetries from the loss functions of neural networks. The method we propose is

---

[*]Equal contribution.
[1]Essentially, this is because the common symmetries in deep learning all contain $\mathbb{Z}_2$ as a subgroup.

theoretically justified and only takes one line of code to implement. By removing these symmetries, our method allows neural networks to explore a more diverse set of parameter spaces and access more expressive solutions. The main contributions of this work are:

1. We show how discrete symmetries in the model can severely limit the expressivity of neural networks in the form commonly known as "collapses" (Section 4);
2. We propose a simple method that provably removes almost all symmetries in neural networks, without having any knowledge of the symmetry (Section 5);
3. We apply the method to solve a broad range of practical problems where symmetry-impaired training can be a major concern (Section 6).

We introduce the notations and problem setting for our work in the next section. Closely related works are discussed in Section 3. All proofs and experimental details are deferred to the appendix.

## 2 DISCRETE SYMMETRIES IN NEURAL NETWORKS

**Notation.** For a matrix $A$, we use $A^+$ to denote the pseudo-inverse of $A$. For groups $U$ and $G$, $U \triangleleft G$ denotes that $U$ is a subgroup of $G$. For a vector $\theta$ and matrix $D$, $\|\theta\|_D^2 := \theta^T D \theta$ is the norm of $\theta$ with respect to $D$. $\odot$ denotes the element-wise product between vectors.

Let $f(\theta, x)$ be a function of the model parameters $\theta$ and input data point $x$. For example, $f$ could either be a sample-wise loss function $\ell$ or the model itself. Whenever $f$ satisfies the following condition, we say that $f$ has the $P$-reflection symmetry (general symmetry groups are dealt with in Theorem 5 in Section 5).

**Definition 1.** *Let $P$ be a projection matrix and $\theta'$ be a point. $f$ is said to have the $(\theta', P)$-reflection symmetry if for all $x$ and $\theta$, (1) $f(\theta + \theta', x) = f((I - 2P)\theta + \theta', x)$, and (2) $P\theta' = \theta'$.*

The second condition is due to the fact that there is a redundancy in the choice of $\theta'$ when $\theta' \neq 0$. Requiring $P\theta' = \theta'$ removes this redundancy and makes the choice of $\theta'$ unique. Since every projection matrix can be written as a product of a (full-rank or low-rank) matrix $O$ with orthonormal columns, one can write $P = OO^T$ and refer to this symmetry as an $O$ symmetry. In common deep learning scenarios, it is almost always the case that $\theta' = 0$ (for example, this holds for the common cases of rescaling symmetries, (double) rotation symmetries, and permutation symmetries, see Theorem 2-4 of Ziyin (2024)). A consequence of $\theta' = 0$ is that the symmetric projection $P\theta$ of any $\theta$ always has a smaller norm than $\theta$: thus, a symmetric solution is coupled to the solutions of weight decay, which also favors small-norm solutions. As an example of reflection symmetry, consider a simple tanh network $f(\theta, x) = \theta_1 \tanh(\theta_2 x)$. The model output is invariant to a simultaneous sign flip of $\theta_1$ and $\theta_2$. This corresponds to a reflection symmetry whose projection matrix is the identity $P = ((1, 0), (0, 1))$. The symmetric solutions correspond to the trivial state where $\theta_1 = \theta_2 = 0$.

To be more general, we allow $\theta'$ to be nonzero to generalize the theory and method to generic hyperplanes since the purpose of the method is to remove all reflection symmetries that may be hidden or difficult to enumerate. Let us first establish some basic properties of a loss function with $P$-reflection, to gain some intuition (again, proofs are in the appendix).

**Proposition 1.** *Let $f$ have the $(\theta', P)$-symmetry and let $f'(\theta) = f(\theta + \theta^\dagger)$. Then, (1) for any $\theta^\dagger$ such that $P\theta^\dagger = \theta'$, $f(\theta + \theta^\dagger) = f((I - 2P)\theta + \theta^\dagger)$, and (2) $f'$ has the $(\theta' - \theta^\dagger, P)$ symmetry.*

Therefore, requiring $P\theta' = \theta'$ is without loss of generality: it only reduces different manifestations of the symmetry to a unique one and simply shifting the function will not remove any symmetry. This proposition emphasizes the subtle difficulty in removing a symmetry.

## 3 RELATED WORKS

One closely related work is that of Ziyin (2024), which shows that every reflection symmetry in the model leads to a low-capacity solution that is favored when weight decay is used. This is because the minimizer of the weight decay is coupled with stationary points of the reflection symmetries – the projection of any parameter to a symmetric subspace always decreases the norm of the parameter, and is thus energetically preferred by weight decay. Our work develops a method to decouple symmetries and weight decay, thus avoiding collapsing into low-capacity states during training. Besides weight decay, an alternative mechanism for this type of capacity loss is gradient-noise induced collapse, which happens when the learning rate - batchsize ratio is high (Chen et al., 2023).

Contemporarily, Lim et al. (2024) empirically explores how removing symmetries can benefit neural network training and suggests a heuristic for removing symmetries by hold a fraction of the weights unchanged during training. However, the proposed method is only proved to work for explicit permutation symmetries in fully connected layers. This is particularly a problem because most of the symmetries in nonlinear systems are unknown and hidden (Cariglia, 2014; Frolov et al., 2017). In sharp contrast, the technique proposed in our work is both architecture-independent and symmetry-agnostic and provably removes all known and unknown discrete symmetries in the loss.

## 4    SYMMETRY IMPAIRS MODEL CAPACITY

We first show that reflection symmetry directly affects the model capacity. For simplicity, we let $\theta' = 0$ for all symmetries. Let $f(x, \theta) \in \mathbb{R}$ be a Taylor-expandable model that contains a $P$-reflection symmetry. Let $\Delta = \theta - \theta_0$. Then, close to any symmetric point $\theta_0$ (any $\theta_0$ for which $P\theta_0 = 0$), for all $x$, Ziyin (2024) showed that

$$f(x, \theta) - f(x, \theta_0) = \underbrace{\nabla_\theta f(x, \theta_0)(I - P)\Delta + O(\|P\Delta\|)^2}_{\text{Symmetry subspace}} + \underbrace{\frac{1}{2}\Delta^T P H(x) P \Delta + O(\|\Delta\|^3)}_{\text{Symmetry-broken subspace}}, \quad (2)$$

where $H(x)$ is the Hessian matrix of $f$. An important feature is that the symmetry subspace is a generic expansion where both odd and even terms are present, and the first order term does not vanish in general. In contrast, in the symmetry-broken subspace, all odd-order terms in the expansion vanish, and the leading order term is the second order. This implies that close to a symmetric solution, escaping from it will be slow, and if at the symmetric solution, it is impossible for gradient descent to leave it. The effect of this entrapment can be quantified by the following two propositions.

**Proposition 2.** *(Symmetry removes feature.) Let $f$ have the $P$-symmetry, and $\theta$ be intialized at $\theta_0$ such that $P\theta_0 = 0$. Then, the kernalized model, $g(x, \theta) = \lim_{\lambda \to 0}(\lambda^{-1} f(x, \lambda\theta + \theta_0) - f(x, \theta_0))$, converges to*

$$\theta^* = A^+ \sum_x \nabla_\theta f(x, \theta_0)^T y(x) \quad (3)$$

*under GD for a sufficiently small learning rate. Here $A := (I - P) \sum_x \nabla_\theta f(x, \theta_0)^T \nabla_\theta f(x, \theta_0)(I - P)$ and $A^+$ denotes the Moore–Penrose inverse of $A$.*

This means that in the kernel regime[2], being at a symmetric solution implies that the feature kernel features are being masked by the projection matrix:

$$\nabla_\theta f(x, \theta_0) \to (I - P)\nabla_\theta f(x, \theta_0), \quad (4)$$

and learning can only happen given these masks. This implies that the model is not using the full feature space that is available to it.

**Proposition 3.** *(Symmetry reduces parameter dimension.) Let $f$ have the $P$-symmetry, and $\theta \in \mathbb{R}^d$ be intialized at $\theta_0$ such that $P\theta_0 = 0$. Then, for all time steps $t$ under GD or SGD, there exists a model $f'(x, \theta')$ and sequence of parameters $\theta'_t$ such that for all $x$,*

$$f'(x, \theta'_t) = f(x, \theta_t), \quad (5)$$

*where $\dim(\theta') = d - \text{rank}(P)$.*

The existence of this type of dimension reduction when symmetry is present has also been noticed by previous works in case of permutation symmetry (Simsek et al.,

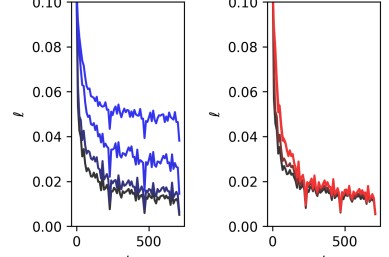

Figure 1: Training loss $\ell$ as a function of the iteration $t$ for a fully connected network on MNIST. Starting from a low-capacity state, vanilla neural networks are trapped under SGD or Adam training (**left**). Blacker lines correspond to higher-capacity initializations, where more neurons are away from the permutation-symmetric state. When the symmetries are removed, the capacity of the initialization no longer affects the solution found at the end of the training (**right**).

2021). Intuitively, the above two results follow from the fact that the symmetric subspaces of reflection symmetries (and any general discrete symmetries) are a linear subspace, and so gradient descent (and thus gradient flow) cannot take the model away from it, despite the discretization error. This means that, essentially, the model is identical to a model with a strictly smaller dimension

---

[2]Technically, this is the lazy training limit (Chizat et al., 2018).

throughout its training, no matter whether the training is through GD or SGD. When there are multiple symmetries, there is a compounding effect. See Figure 1 for an illustration. We initialize a two-layer ReLU neural network on a low-capacity state where a fraction of the hidden neurons are identical (corresponding to the symmetric states of the permutation symmetry) and train with and without removing the symmetries. We see that when the symmetries are removed (with the method proposed in the next section), the model is no longer stuck at these neuron-collapsed solutions. Also, see Section C for the evolution of NTK and gradient covariance during training.

## 5 REMOVING SYMMETRY WITH STATIC BIAS

Next, we prove that a simple algorithm that involves almost no modification to any deep learning training pipeline can remove almost all such symmetries from the loss without creating new ones. From this section onward, we will consider the case where the function under consideration is the loss function (a per-batch loss or its expectation): $f = \ell$.

### 5.1 COUNTABLE SYMMETRIES

We seek an algorithm that eliminates the reflection symmetries from the loss function $\ell$. This subsection will show that when the number of reflection symmetries in the loss function is countable, one can completely remove them using a simple technique. The symmetries are required to have the following property and the loss function is assumed to obey assumption 1.

**Property 1.** *(Enumeratability) There exists a countable set of pairs of projection matrices and biases $S = \{(\theta_i^\dagger, P_i)\}_i^N$ such that $\ell(\theta, x)$ has the $\theta_i^\dagger$-centric $P_i$-reflection symmetry for all $i$. In addition, $\ell$ does not have any $(\theta^\dagger, P)$ symmetry for $(\theta, P) \notin S$.*

**Assumption 1.** *There only exists countably many pairs $(c_0, \tilde{\theta})$ such that $g(x) = \ell(\theta, x) - c_0\theta$ contains a $\tilde{\theta}$-centric $P$ symmetry, where we require $Pc_0 = c_0$ and $P\tilde{\theta} = \tilde{\theta}$.*

This assumption is satisfied by common neural networks with standard activations. The main purpose of this assumption is to rule out the pathological of a linear or quadratic deterministic objective, which never appears in practice or for which symmetry is not a concern.[3]

For symmetry removal, we propose to utilize the following alternative loss function. Let $\theta_0$ be drawn from a Gaussian distribution with variance $\sigma_0$ and $\ell$ be the original loss function:

$$\ell_{\mathrm{r}}(\theta, x) = \ell(\theta + \theta_0) + \gamma\|\theta\|^2. \tag{6}$$

$\gamma$ is nothing but the standard weight decay. We will see that using a static bias along with weight decay is essential for the method to work. We find that with unit probability, the loss function $\ell_{\mathrm{r}}$ contains no reflection symmetry:

**Theorem 1.** *Let $\ell$ satisfy Property 1 and Assumption 1. Then, with probability $1$ over the sampling of $\theta_0$, there exists no projection matrix $P$ and reflection point $\theta'$ such that $\ell_{\mathrm{r}}$ has the $(\theta', P)$-symmetry.*

The core mechanism of this theorem is decoupling of the solutions of the symmetries from the solutions of the weight decay. With weight decay, a solution with a small norm is favored, whereas with a random bias, the symmetric solutions are shifted by a small constant and no longer overlap with solutions that have a small norm. Equivalently, this method can be implemented as a random bias in the $L_2$ regularization: $\ell_{\mathrm{r}}(\theta, x) = \ell(\theta) + \gamma\|\theta + \theta_0\|^2$. This is a result of the fact that simple translation does not change the nature of the function. There are two remarkable parts of the theorem: (1) it not only removes all existing symmetries but also guarantees that there are no remaining ones; (2) the proof works as long as the variance of $\theta_0$ is nonzero. This hints at the possibility of using a small and random $\theta_0$, which removes all symmetries in principle and also does not essentially affect the solutions of the original objective. In this work, we will refer to the method in Eq. (6) as *syre*, an abbreviation for "symmetry removal."

### 5.2 UNCOUNTABLY MANY SYMMETRIES

In deep learning, it is possible for the model to simultaneously contain infinitely many reflection symmetries. This happens, for example, when the model parameters have the rotation symmetry or the double rotation symmetry (common in self-supervised learning problems Ziyin et al. (2023) or

---

[3]An example that violates this assumption is when $\ell(\theta, x) = c_0^T\theta$. $\ell(\theta, x) - c_0^T\theta$ has infinitely many reflection symmetries everywhere and for every data point because it is a constant function.

transformers). It turns out that adding a static bias and weight decay is not sufficient to remove all symmetries, but we will show that a simple modification would suffice.

We propose to train on the following alternative loss instead, where $ar$ stands for "advanced removal":

$$\ell_{\mathrm{ar}}(\theta) = \ell(\theta + \theta_0) + \gamma\|\theta\|_D^2, \tag{7}$$

where $D$ is a positive diagonal matrix in which all diagonal elements of $D$ are different. The simplest way to achieve such a $D$ is to set $D_{ii} \sim \mathrm{Uniform}(1 - \epsilon, 1 + \epsilon)$, where $\epsilon$ is a small quantity.

**Theorem 2.** *Any $(\theta', P)$-symmetry that $\ell_{\mathrm{ar}}$ satisfies obeys: (1) $P\theta_0 = \theta'$ (2) and $PD = DP$.*

Conditions (1) and (2) implies that there are at most finitely many $(\theta', P)$-symmetry $\ell_{\mathrm{ar}}$ can have. When there does not exist any symmetry that satisfies this condition, we have removed all the symmetries. In the worst case where $\ell$ is a constant function, there are $2^N$ symmetries where $N$ is the number of reflection symmetries. If we further assume that every $P$ is associated with at most finitely many $\theta'$, then we, again, remove all symmetries with probability $1$. The easiest way to determine this $D$ matrix is through sampling from a uniform distribution with a variance $\sigma_D \ll 1$.

## 5.3 STRENGTH OF SYMMETRY REMOVAL

While any level of $\theta_0$ and $\sigma_D$ are sufficient to remove the symmetries, one might want to quantify the degree to which the symmetries are broken. This is especially relevant when the model is located close to a symmetric solution and requires a large gradient to escape from it. Also, a related question that may arise in practice is how large one should choose $\sigma_0$ and $\sigma_D$, which are the variances of $\theta_0$ and $D$. The following theorem gives a quantitative characterization of the degree of symmetry breaking.[4]

**Theorem 3.** *Let the original loss satisfy a $(\theta^*, P)$-symmetry, where $\theta^* \in \mathbb{R}^d$. Then, for any local minimum $\theta \in \Theta(1)$, if $\sigma_D = o(\sigma_0)$ and $P\theta \neq 0$,*

$$\Delta := \frac{1}{\|P\theta\|}\left[\ell_{\mathrm{ar}}(\theta + \theta^*) - \ell_{\mathrm{ar}}((I - 2P)\theta + \theta^*)\right] = \Omega(\gamma\sigma_0). \tag{8}$$

This theorem essentially shows a "super-Lipschitz" property of the difference between the loss function values between parameters and their reflections. This means that with a random bias, the symmetry will be quite strongly removed, as long as we ensure $\sigma_D \ll \sigma_0$, which is certainly ensured when $\sigma_D = 0$. As a corollary, it also shows that after applying a static bias, no symmetric solution where $P\theta = 0$ can still be a stationary point because as $P\theta \to 0$, the quantity $\Delta$ converges to the projection of the gradient onto symmetry breaking subspace.

**Corollary 1.** *For any $\theta$ such that $P\theta = 0$, $P\nabla_\theta\ell_{\mathrm{ar}} = \Omega(\gamma\sigma_0)$.*

Now, the more advanced question is the case when there are multiple reflection symmetries, and one wants to significantly remove every one of them.

**Theorem 4.** *Let $\ell$ contain $N$ reflection symmetries: $\{(P_i, \theta_i^*)\}_{i=1}^N$. Let*

$$\Delta_i := \frac{\ell_{\mathrm{ar}}(\theta + \theta_i^*) - \ell_{\mathrm{ar}}((I - 2P_i)\theta + \theta_i^*)}{\|P_i\theta\|}. \tag{9}$$

*Then, for any local minimum $\theta \in \Theta(1)$, letting $\gamma\sigma_0 = \Omega\left(\frac{2\epsilon N}{1-\delta}\right)$ guarantees that $\Pr(\min_i |\Delta_i| > \epsilon) > \delta$ for any $\epsilon$ and $\delta < 1$.*

In the theorem, the probability is taken over the random sampling of the static bias. Namely, the achievable strengths of symmetry-breaking scales inversely linearly in $N$, the size of the minimal set of the entire group generated by $N$ reflections. In general, without further assumptions there is no way to improve this scaling because, for example, the smallest of $N$ independent bounded variables roughly scales as $1/N$ towards its lower boundary.

**General Groups.** Lastly, one can generalize the theory to prove that the proposed method removes symmetries from a generic group. Let $G$ be the linear representation of a generic finite group, possibly with many nontrivial subgroups. If the loss function $\ell$ is invariant under transformation by the group $G$, then

$$\forall g \in G, \ \ell(\theta) = \ell(g\theta). \tag{10}$$

---

[4]Because $\theta_0$ and $D$ are randomly sampled, the big-$O$ notation $x \in O(z)$ is used to mean that as the scaling parameter tends to infinity, $\exists c_0$ such that $\Pr(\|x\| < c_0 z) \to 1$.

Because $G$ is finite, it follows that the representations $g$ must be full-rank and unipotent.

The following theorem shows that at every symmetric solution, there exists an escape direction with a strong gradient that pulls the parameters away from every subgroup of the related symmetry. In other words, it is no longer possible to be trapped in a symmetric solution. While general groups appear less common, they exist in a lot of application scenarios with equivariant networks (Cohen & Welling, 2016), where it is common to incorporate group structures into the model.

Letting $U$ be a subgroup of $G$, we denote with the overbar the following matrix:

$$\overline{U} = \frac{1}{|U|} \sum_{u \in U} u. \tag{11}$$

Note that $\overline{U}$ is a projection matrix: $\overline{U}\,\overline{U} = \overline{U}$. This means that $I - \overline{U}$ is also a projection matrix. Importantly, $\overline{U}$ projects a vector into the symmetric subspace. For any $u \in U$, $u\overline{U} = \overline{U}$. Likewise, $I - \overline{U}$ projects any vector into the symmetry-broken subspace, a well-known result in the theory of finite groups (Gorenstein, 2007). We denote by

$$\Delta_V = \|(I - \overline{V})\nabla_\theta \ell\| \tag{12}$$

the strength of the symmetry removal for the subgroup $V$.

**Theorem 5.** *Let $\Gamma(G)$ denote the smallest minimal generating set for the group $G$. $Z$ denotes the number of minimal subgroups of $G$. Let $\ell$ be invariant under the group transformation $G$ and let $\theta$ be in the invariant subspace of a subgroup $U \triangleleft G$. Then, for every subgroup $V \triangleleft U \triangleleft G$,*
1. *$\Delta_V = \Omega(\gamma\sigma_0 \mathrm{rank}(I - \overline{V}))$;*
2. *$\min_{V \triangleleft U} \Delta_V = \Omega(\gamma\sigma_0 \mathrm{rank}(I - \overline{V})Z^{-1})$;*
3. *if $G$ is abelian, $\min_{V \triangleleft U} \Delta_V = \Omega(\gamma\sigma_0 \mathrm{rank}(I - \overline{V})|\Gamma(U)|^{-1})$;[5]*
4. *additionally, for any $\epsilon > 0$ and $\delta < 1$, $\mathrm{Pr}(\min_{V \triangleleft U} \Delta_V > \epsilon) > \delta$, if $\gamma\sigma_0 = \Omega\left(\frac{2\epsilon|\Gamma(U)|}{1-\delta}\right)$.*

Item (2) of the theorem is essentially due to the fact that removing symmetries from a larger group can be reduced to removing them from one of its subgroups. Conceptually, this result has the same root as the classical theory of combinatorial designs, where averaging over a subset of groups has the same effect as averaging over the whole group (Lindner & Rodger, 2017). In general, $1 \le Z \le |G|$ (and sometimes $\ll |G|$), and so this scaling is not bad. For the part (3) of the theorem, the term $|\Gamma(U)|$ is especially meaningful. It is well-known that $|\Gamma(U)| \le \log|U|$, and so the worst-case symmetry-breaking strength is only of order $1/\log|U|$, which is far slower than what one would expect. In fact, for a finite group with size $N$, the number of subgroups can grow as fast as $N^{\log N}$ (Borovik et al., 1996), and thus, one might naively think that the minimal breaking strength decreases as $N^{-\log N}$. This theorem shows that the proposed method is highly effective at breaking the symmetries in the loss function or the model.

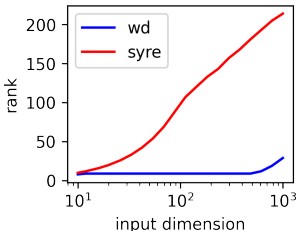

Figure 2: With the proposed method, the rank of the feature covariance matrix increases with the input dimension. The same may happen under vanilla weight decay, but the effect is not strong. The covariance is computed from the first-layer output with batch size 1000, and eigenvalues larger than $10^{-4}$ are truncated to zero.

A numerical example is shown in Figure 2, which validates a major prediction of the theorem: a symmetry is easier to remove if it is high-dimensional. We train a two-layer ReLU net in a teacher-student scenario and change the input dimension. This experiment holds the number of (permutation) symmetries fixed and directly controls $\mathrm{rank}(I - \overline{V})$. As the input dimension increases, the symmetry of the learned model becomes lower. In comparison, without a static bias, having a high dimension is not so helpful.

## 5.4 Hyperparameter and Implementation Remark

As discussed, there are two ways to implement the method (Eq. (6) or Eq. (7)). In our experiments, we stick to the definition of Eq. (6), where the model parameters are biased, and weight decay is the same as the standard implementation. For the choice of hyperparameters, we always set $\sigma_D = 0$

---

[5]This result can be generalized to the case where all projectors $\overline{V}$ commute with each other, even if $G$ is nonabelian. Namely, this is a consequence of the properties of the representations of $G$ and of $G$ itself.

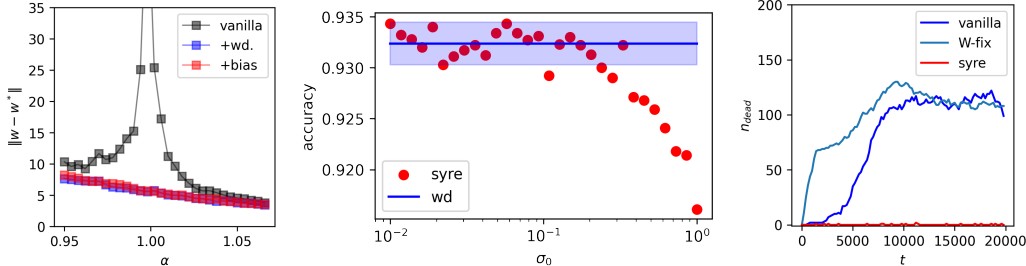

Figure 3: Training with syre in standard settings. The result shows that biasing the models by a small static bias does not change the performance of standard training settings. **Left**: Application of the method to a linear regression problem. Here, $\alpha^{-1} = d/N$ is the degree of parameterization. A well-known use of weight decay is to prevent double descent when $\alpha = 1$. Here, we see that the proposed method works as well as vanilla weight decay. Because there is no reflection symmetry in the problem, the proposed method should approximate vanilla weight decay. **Mid**: Test accuracy of Resnet18 on the CIFAR-10 datasets. The blue line denotes the performance of Resnet, and the shadowed area denotes its standard deviation estimated over 10 trials. For $\sigma_0 < 0.2$, there is no significant difference between the performance of the vanilla Resnet and *syre* Resnet. **Right**: linear regression with a redundant parametrization (Poon & Peyré, 2021). The loss function takes the form $\ell(u, w) = \|(u \odot w)^T x - y\|^2$. Due to symmetry, the point $(u_i, w_i) = 0$ is a low-capacity state where the $i$-th neuron is "dead". Training with style, the model stayed away from any trapping low-capacity state during training. In comparison, training with vanilla SGD or a heuristic for fixing the weights does not fix the problem of collapsing to a low-capacity state.

as we find only introducing $\sigma_0$ to be sufficient for most tasks. Experiments with standard training settings (see the next section for the Resnet18 experiment on CIFAR-10) show that choosing $\sigma_0$ to be at least an order of magnitude smaller than the standard initialization scale (usually of order $1/\sqrt{d}$ for a width of $d$) works the best. We thus recommend a default value of $\sigma_0$ to be $0.01/\sqrt{d}$, where $1/d$ is the common initialization variance. For the rest of the paper, we state $\sigma_0$ in relative units of $\sqrt{d}^{-1}$ for this reason. That being said, we stress that $\sigma_0$ is a hyperparameter worth tuning, as it directly controls the tradeoff between optimization and symmetry removal.

## 6 EXPERIMENT

First, we show that the proposed method is compatible with standard training methods. We then apply the method to a few settings where symmetry is known to be a major problem in training. We see that removing symmetries with the proposed method is well correlated with improved model performance for these problems. Additional experiments are presented in Appendix B, including the time and memory usage of *syre* and its application to transformers and improving mode connectivity.[6]

### 6.1 COMPATIBILITY WITH STANDARD TRAINING

**Ridge linear regression.** Let us first consider the classical problem of linear regression with $d$-dimensional data, where one wants to find $\min_w \sum_i (w^T x_i - y_i)^2$. Here, the use of weight decay has a well-known effect of preventing the divergence of generalization loss at a critical dataset size $N = d$ (Krogh & Hertz, 1992; Hastie et al., 2019). This is due to the fact that the Hessian matrix of the loss becomes singular exactly at $N = d$ (at infinite $N$ and $d$). The use of weight decay shifts all the eigenvalues of the Hessian by $\gamma$ and removes this singularity. In this case, one can show that the proposed method is essentially identical to the simple ridge regression. The ridge solution is $w^* = \mathbb{E}[\gamma I + A]^{-1}\mathbb{E}[xy]$, where $A = \mathbb{E}[xx^T]$, and the solution to the biased model is

$$w^* = \mathbb{E}[\gamma I + A]^{-1}(\mathbb{E}[xy] + \gamma\theta_0). \tag{13}$$

The difference is negligible with the original solution if either $\gamma$ and $\theta_0$ are small. See Figure 3-left.

**Reparametrized Linear Regression.** A minimal model with emergent interest in the problem of compressing neural networks is the reparametrized version of linear regression (Poon & Peyré, 2021; Ziyin & Wang, 2023), the loss function of which is $\ell(u, w) = \|(u \odot w)^T x - y\|^2$, where we let $u, w, x \in \mathbb{R}^{200}$ and $y \in \mathbb{R}$. Due to the rescaling symmetry between every parameter $u_i$ and $w_i$, the solutions where $u_i = w_i = 0$ is a low-capacity state where the $i$-th neuron is "dead."

---

[6]An implementation of *syre* can be found at https://github.com/xu-yz19/syre/.

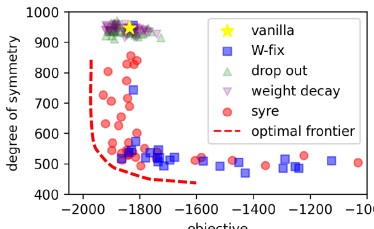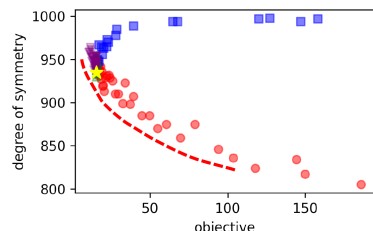

Figure 4: The degree of symmetry versus the objective value for two choices of $B$ and various training methods with different hyperparameters. The proposed method is the only method to smoothly interpolate between optimized solutions and solutions with low symmetry. *syre* performs well in both cases. **Left**: objective with unstructured symmetry **Right**: structured symmetry.

For this problem, we compare the training with standard SGD and *syre*. We also compare with a heuristic method (**W-fix**), where a fraction $\phi = 0.3$ of weights of every layer is held fixed with a fixed variance $\kappa = 0.01$ at initialization. This method has been suggested in Lim et al. (2024) as a heuristic for removing symmetries and is found to work well when there is permutation symmetry. We see that both the vanilla training and the W-fix collapse to low-capacity states during training, whereas the proposed method stayed away from them throughout. The reason is that the proposed method is model-independent and symmetry-agnostic, working effectively for any type of possibly unknown symmetry in an arbitrary architecture.

**ResNet.** We also benchmark the performance of the proposed method for ResNet18 with different $\sigma_0$ on the CIFAR-10 datasets in Figure 3. When $\sigma_0 = 0$, the *syre* model is equivalent to the vanilla model. Figure 3 shows that the difference in performance between the vanilla Resnet and *syre* Resnet is very small and becomes neglectable when $\sigma_0 < 0.2$.

## 6.2 BENCHMARKING SYMMETRY REMOVAL

In this section, we benchmark the effect of symmetry control of the proposed method for two controlled experiments. To compare the influence of *syre* and other training methods on the degree of symmetry, we consider minimizing the following objective function:

$$(w^T w)^2 - w^T B w := (w^T w)^2 - \sum_{i=1}^{d} \lambda_i (v_i^T w)^2,$$

where $w \in \mathbb{R}^d$ is the optimization parameter and $B \in \mathbb{R}^{d \times d}$ is a given symmetric matrix with eigenvalues $\lambda_i$ and eigenvectors $v_i$ ($v_i^T v_i = 1$). The objective function has $n$ reflection symmetries $P_i w := w - 2(v_i^T w) v_i$. Hence, we define the degree of symmetry as $\sum_{i=1}^{d} \mathbf{1}\{v_i^T w < c_{\text{th}}\}$, where $c_{\text{th}}$ is a given threshold. Depending on the spectrum of $B$, the nature of the task is different. We thus consider two types of spectra: (1) an unstructured spectrum where $B = G + G^T$ for a Gaussian matrix $G$, and (2) a structured spectrum where $B = \text{diag}(v)$ where $v$ is a random Gaussian vector. Conceptually, the first type is more similar to rotation and double rotation symmetries in neural networks where the basis can be arbitrary, while the second is a good model for common discrete symmetries where the basis is often diagonal or sparse. For the first case we choose $c_{\text{th}} = 10^{-3}$ and for the second case we choose $c_{\text{th}} = 10^{-1}$.

In Figure 4, we compare *syre*, W-fix, drop out, weight decay, and the standard training methods in this setting for $d = 1000$ and two choices of $B$. In both cases, we use Gaussian initialization and gradient descent with a learning rate of $10^{-4}$. For *syre* and weight decay, we choose weight decay from 0.1 to 10. For W-fix, we choose $\phi$ from 0.001 to 0.1. For dropout, we choose a dropout rate from 0.01 to 0.6. Figure 4 shows that for both cases, *syre* is the only method that effectively and smoothly interpolates between solutions with low symmetry and best optimization. This is a strong piece of evidence that the proposed method can *control* the degree of symmetries in the model.

## 6.3 FEATURE AND NEURON COLLAPSES IN SUPERVISED LEARNING

See Figure 5, where we train the vanilla and *syre* four-layer networks with various levels of weight decay $\gamma$ and input-output covariance $\alpha$. The dataset is constructed by rescaling the input by a factor of $\alpha$ for the MNIST dataset. The theory predicts that the *syre* model can remove the permutation symmetry in the hidden layer. This is supported by subfigures in Figure 5, where vanilla training results in a low-rank solution. Meanwhile, the accuracy of the low-rank solution is significantly lower for a large $\gamma$ or a small $\alpha$, which corresponds to the so-called neural collapses. Also, we

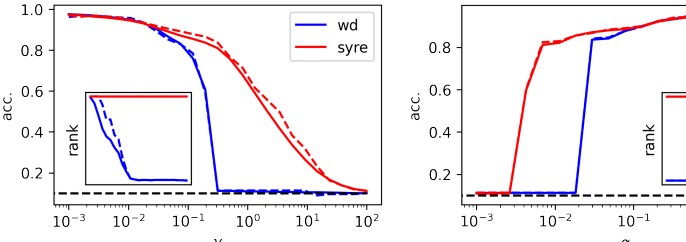

Figure 5: Performance of a 4-layer FCN for datasets with various weight decay $\gamma$ and data distributions with varyings strengths of linear correlation $\alpha$. As the theory predicts, the covariance of the model with vanilla weight decay has a low-rank structure and performs significantly worse. In the main figures, solid lines denote training accuracy and dashed lines denote test accuracy. The dashed black line corresponds to random guess. Subfigures show the rank of the covariance matrix of the first layer output before (solid lines) and after (dashed lines) activation with batch size 1000. We set eigenvalues smaller than $10^{-4}$ to 0. **Left**: $\alpha = 1$ and different $\gamma$. **Right**: $\gamma = 0.01$ and different $\alpha$. A similar experiment for vision transformers is presented in Section B.5.

| | Hyperparameter | Low-rankness | Penult. Layer Acc. | Last Layer Acc. |
|---|---|---|---|---|
| vanilla | - | 70% | 46.8% | 22.2% |
| *syre* | $\sigma_0 = 0.1$ | 0% | 46.8% | 31.7% |
| | $\sigma_0 = 0.01$ | 0% | 46.2% | 32.5% |
| | $\sigma_0 = 0.1$, all layers | 0% | 44.6% | 30.7% |
| | $\sigma_0 = 0.01$, all layers | 0% | 45.4% | 32.4% |

Table 1: Performance of the linearly evaluated Resnet18 on CIFAR100 for the unsupervised self-constrastive learning task. Here, the low-rankness measures the proportion of eigenvalues smaller than $10^{-5}$. Our experiment indicates that symmetry-induced reduction in model capacity can explain about 50% of the performance difference between the representation of the two layers.

observe that *syre* shifts the eigenvalues of the representation by a magnitude proportional to $\sigma_0$, thus explaining the robustness of the method against collapses in the latent representation (See Figure 8).

## 6.4 POSTERIOR COLLAPSE IN BAYESIAN LEARNING

Wang & Ziyin (2022) points out that a type of posterior collapse in Bayesian learning (Lucas et al., 2019; Wang et al., 2021) is caused by the low-rankness of the solutions. We show that training with *syre* could overcome this kind of posterior collapse. In Figure 9, we train a $\beta$-VAE (Kingma & Welling, 2013; Higgins et al., 2016) on the Fashion MNIST dataset. Following Wang & Ziyin (2022), we use $\beta$ to weigh the KL loss, which can be regarded as the strength of prior matching. Both the encoder and the decoder are a two-layer network with SiLU activation. The hidden dimension and the latent dimension are 200. Only the encoder has weight decay because the low-rank problem is caused by the encoder rather than the decoder. We also choose the prior variance of the latent variable to be $\eta_{enc} = 0.01$. Other settings are the same as Wang & Ziyin (2022). Posterior collapse happens at $\beta = 10$, signalized by a large reconstruction loss in the right side of Figure 9. However, the reconstruction loss decreases, and the rank of the encoder output increases (according to the left side of Figure 9) after we use weight decay and *syre*. This is further verified by the generated image in Figure 10. Therefore, we successfully remove the permutation symmetry of the encoder.

## 6.5 LOW-CAPACITY TRAP IN SELF-SUPERVISED LEARNING

A common but bizarre practice in self-supervised learning (SSL) is to throw away the last layer of the trained model and use the penultimate learning representation, which is found to have much better expressiveness than the last layer representation. From the perspective of symmetry, this problem is caused by the rotation symmetry of the last weight matrix in the SimCLR loss. We train a Resnet-18 together with a two-layer projection head over the CIFAR-100 dataset according to the setting for training SimCLR in Chen et al. (2020). Then, a linear classifier is trained using the learned representations. Our implementation reproduces the typical accuracy of SimCLR over the CIFAR-100 dataset (Patacchiola & Storkey, 2020). As in Chen et al. (2020), the hidden layer before the projection head is found to be a better representation than the layer after. Therefore, we apply our *syre* method to the projection head or to all layers. According to Table 1, *syre* removes the low-rankness of the learned features and increases the accuracy trained with the features after projection while not

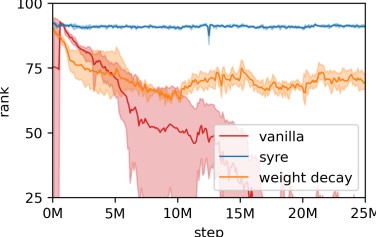 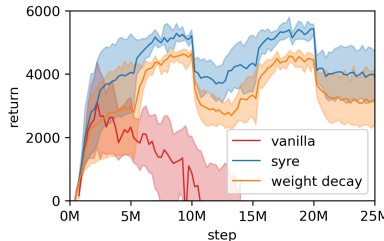

Figure 6: Loss of plasticity in continual learning in an RL setting. We use the PPO algorithm (Schulman et al., 2017) to solve the Slippery-Ant problem (Dohare et al., 2023). The rank and the performance of the vanilla PPO decrease quickly, while the rank and the performance of PPO with *syre* remain the same, beyond that of PPO with weight decay. **Left**: the effective rank of the policy network as defined in Dohare et al. (2023). **Right**: returns. Each trajectory is averaged over 5 different random seeds.

changing the representation before projection. Thus, symmetry-induced reduction in model capacity can explain about $50\%$ of the performance difference between the representation of the two layers. Also, an interesting observation is that just improving the expressivity of the last layer is insufficient to close the gap between the performance of the last layer and the penultimate layer. This helps us gain a new insight: symmetry is not the only reason why the last layer representation is defective.

## 6.6 LOSS OF PLASTICITY IN CONTINUAL LEARNING

A form of low-capacity collapse also happens during continual learning, i.e., the plasticity of the network gradually decreases as the model is trained on more and more tasks. This problem is common in both supervised and reinforcement learning settings and may also be relevant to the finetuning of large language models (McCloskey & Cohen, 1989; Ash & Adams, 2020; Dohare et al., 2021; Abbas et al., 2023).

**Supervised Learning.** In Figure 11, we train a CNN with two convolution layers (10 channels and 20 channels) and two fully connected layers (320 units and 50 units) over the MNIST datasets. For the data, we randomly permute the pixels of the training and test sets for 9 times, forming 10 different tasks (including the original MNIST). We then train a vanilla CNN and a *syre* CNN over the 10 tasks continually with SGD and weight decay 0.01. The inset of Figure 11 shows that the rank of the original model gradually decreases, but the *syre* model remains close to full rank. Correspondingly, in the right side of Figure 11, the accuracy over the test set drops while the rank of the original model collapses, but the accuracy of the *syre* model remains similar.

**Reinforcement Learning.** In Figure 6, we use the PPO algorithm for the Slippery-Ant problem (Dohare et al., 2023), a continual variant of the Pybullet's Ant problem (Coumans & Bai, 2016) with friction that changes every $5M$ steps. Hyperparameters for the PPO algorithm are borrowed from Dohare et al. (2023), and we use a weight decay of 0.002 for both PPO with weight decay and with *syre*. Figure 6 suggests that *syre* is effective in maintaining the rank of the model during continual training and obtains better performance than pure weight decay.

## 7 CONCLUSION

Symmetry-induced neural-network training problems exist extensively in machine learning. We have shown that the existence of symmetries in the model or loss function may severely limit the expressivity of the trained model. We then developed a theory that leverages the power of representation theory to show that adding random static biases to the model, along with weight decay, is sufficient to remove almost all symmetries, explicit or hidden. We have demonstrated the relevance of the method to a broad range of applications in deep learning, and a possible future direction is to deploy the method in large language models, which naturally contain many symmetries. Lastly, it is worth noting that on its own, symmetry is neither good nor bad. For example, practitioners may be interested in introducing symmetries to the model architecture in order to control the capacity or hierarchy of the model (Ziyin et al., 2025). However, with too much symmetry, the training of models becomes slow and likely to contain many low-capacity traps. Meanwhile, a model completely without symmetry may have undesirably high capacity and be more prone to overfitting. Having the right degree of symmetry might thus be crucial for achieving both smooth optimization and good generalization. With our proposed method, it becomes increasingly possible to deliberately fine-grain engineer symmetries in the loss function, introducing desired symmetries and removing undesirable ones.

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

# A THEORETICAL CONCERNS

## A.1 PROOF OF PROPOSITION 1

*Proof.* Part (1). Note that we have

$$\theta^\dagger = P\theta^\dagger + (I - P)\theta^\dagger = \theta' + (I - P)\theta^\dagger. \tag{14}$$

Thus,

$$(I - 2P)(I - P)\theta^\dagger = (I - P)\theta^\dagger. \tag{15}$$

Therefore, we have

$$f(\theta + \theta^\dagger) = f((\theta + (I - P)\theta^\dagger) + \theta') = f((I - 2P)(\theta + (I - P)\theta^\dagger) + \theta') = f((I - 2P)\theta + \theta^\dagger). \tag{16}$$

This proves part (1).

Part (2). By definition,

$$f'(\theta - \theta^\dagger + \theta') = f(\theta + \theta') \tag{17}$$

$$= f((I - 2P)\theta + \theta') \tag{18}$$

$$= f'((I - 2P)\theta - \theta^\dagger + \theta'). \tag{19}$$

This completes the proof. $\square$

## A.2 PROOF OF PROPOSITION 2

*Proof.* By (2), $g(x, \theta)$ simplifies to a kernel model

$$g(x, \theta) = \nabla_{\theta_0} f(x, \theta_0)(I - P)\theta. \tag{20}$$

Let us consider the squared loss $\ell(\theta) = \sum_x \|y(x) - g(x, \theta)\|^2$ and denote $A := \sum_x (I - P)\nabla_{\theta_0} f(x, \theta_0)^T \nabla_{\theta_0} f(x, \theta_0)(I - P)$, $b := (I - P)\sum_x \nabla_{\theta_0} f(x, \theta_0)^T y(x)$. Assuming the learning rate to be $\eta$, the GD reads

$$\theta^{t+1} = \theta^t - 2\eta(A\theta^t - b), \tag{21}$$

where $\theta^0 = 0$. If

$$\eta < \frac{1}{2\lambda_{max}(A)}, \tag{22}$$

GD converges to

$$\theta^\star = \lim_{t \to \infty} \sum_{k=0}^{t} (I - 2\eta A)^k * 2\eta b \tag{23}$$

$$= A^+ b,$$

which is the well-known least square solution. $\square$

## A.3 PROOF OF PROPOSITION 3

*Proof.* According to Ziyin (2024, Theorem 4), we have

$$P\nabla_\theta \ell(x, \theta_0) = 0. \tag{24}$$

Therefore, after one step of GD or SGD, we still have $P\theta_1 = 0$. By induction, we have $P\theta_t = 0$.

Finally, suppose that $\{a_i\}_{i=1}^{d-\text{rank}(P)}$ forms a basis of $\text{ker}P$, and define $f'(x, \theta') := f(x, \sum_{i=1}^{d-\text{rank}(P)} \theta'_i a_i)$ for $\dim(\theta') = d - \text{rank}(P)$. By choosing $\theta'_i = \theta^T a_i$, we have $f'(x, \theta'_t) = f(x, \theta_t)$. $\square$

## A.4 LEMMAS

**Lemma 1.** *Let $x \in \mathbb{R}^d$ and $P$ be a projection matrix. Let $f(x)$ be a scalar function that satisfies*

$$f(x + x') = f((I - 2P)x + x') + c^T Px, \tag{25}$$

*where $c$ is a constant vector. Then, there exists a unique function $g(x)$ such that*

    1. *$g(x)$ has the $x'$-centric $P$-symmetry,*

    2. *and $f(x) = g(x) + \frac{1}{2} c_0^T Px$.*

*Proof.* (a) Existence. $f(x) = g(x) + \frac{1}{2} c_0^T x$. Let us suppose $g(x)$ is not $x'$-centric $P$-symmetry. Then, there exists $x$ such that

$$g(x + x') - g((I - 2P)x + x') = \Delta \neq 0. \tag{26}$$

Then, by definition, we have that

$$c_0^T Px = f(x + x') + f((I - 2P)x + x') \tag{27}$$

$$= g(x + x') - \frac{1}{2} c_0^T P(x + x') - g((I - 2P)x + x') - c_0^T P(x + x') \tag{28}$$

$$= \Delta + \frac{1}{2} c_0^T P(x + x') - \frac{1}{2} c_0^T P((I - 2P)x + x') \tag{29}$$

$$= \Delta + c_0^T Px. \tag{30}$$

This is a contradiction. Therefore, there must exist $g(x)$ that satisfies the lemma statement.

(b) Uniqueness. Simply note that the expression of $g$ is uniquely given by[7]

$$g(x) = f(x + x') - f((I - 2P)x + x'). \tag{31}$$

$\square$

## A.5 PROOF OF THEOREM 1

*Proof.* We prove by contradiction. Let us suppose there exists such pair, $(\theta', P)$. By definition, we have that

$$\ell_{\mathrm{r}}(\theta + \theta') = \ell(\theta + \theta' + \theta_0) + \gamma \|\theta + \theta'\|^2. \tag{32}$$

By assumption, we have that

$$\ell((I - 2P)\theta + \theta' + \theta_0) + \gamma \|(I - 2P)\theta + \theta'\|^2 = \ell(\theta + \theta' + \theta_0) + \gamma \|\theta + \theta'\|^2, \tag{33}$$

and, so, for all $\theta$,

$$\ell((I - 2P)\theta + \theta' + \theta_0) = \ell(\theta + \theta' + \theta_0) + 4\gamma \theta^T P\theta'. \tag{34}$$

There are two cases: (1) $P\theta' = 0$ and (2) $P\theta' \neq 0$.

For case (1), we have that $\ell((I - 2P)\theta + \theta' + \theta_0) = \ell(\theta + \theta' + \theta_0)$, but this can only happen if the original loss $\ell$ has the $(\theta' + \theta_0)$-centric $P$-symmetry. By Property 1, this implies that

$$\theta' + \theta_0 = \theta_i^\dagger \tag{35}$$

for some $i$. Applying $P$ on both sides, we obtain that

$$P\theta_0 = P\theta_i^\dagger. \tag{36}$$

But, $\theta_0$ is a random variable with a full-rank covariance while the set $\{P\theta_i^\dagger\}$ has measure zero in the real space, and so this equality holds with probability zero.

---

[7]Alternatively, note that $c^T x$ is odd and that $g(x)$ can be shifted by a constant to be an even function. The uniqueness follows directly from the fact that every function can be uniquely factorized into an odd function and an even function.

For case (2), it follows from Lemma 1 that for a fixed $x$ and $\theta_0$, $\ell(\theta)$ can be uniquely decomposed in the following form

$$\ell(\theta) = g(\theta) - 2\gamma\theta^T P\theta', \tag{37}$$

where $g(\theta)$ has the $\theta' + \theta_0$-centric $P$-symmetry.

Let $c_0 = 2\gamma P\theta'$ and $\tilde{\theta} = P(\theta' + \theta_0)$. Then $\ell(\theta) + c_0\theta$ has the $\tilde{\theta}$-centric $P$ symmetry. We also have $\tilde{\theta} - \frac{c_0}{2\gamma} = P\theta_0$. According to Assumption 1, there are only countable many such $\{c_0, \tilde{\theta}\}$ pairs. However, $P\theta_0$ is a standard Gaussian random variable, which leads to a contradiction with probability 1. $\quad\square$

**Remark.** *It is easy to see that Assumption 1 could be slightly relaxed. We only require $\{\tilde{\theta} - \frac{c_0}{2\gamma}\}$ to have a zero measure, which is also a necessary and sufficient condition.*

## A.6 PROOF OF THEOREM 2

*Proof.* We prove by contradiction. Let us suppose there exists such pair, $(\theta', P)$. By definition, we have that

$$\ell_{ar}(\theta + \theta') = \ell(\theta + \theta' + \theta_0) + \gamma\|\theta + \theta'\|_D^2. \tag{38}$$

By assumption, we have that

$$\ell((I - 2P)\theta + \theta' + \theta_0) + \gamma\|(I - 2P)\theta + \theta'\|_D^2 = \ell(\theta + \theta' + \theta_0) + \gamma\|\theta + \theta'\|_D^2, \tag{39}$$

and, so, for all $\theta$,

$$\ell((I - 2P)\theta + \theta' + \theta_0) = \ell(\theta + \theta' + \theta_0) + 4\theta^T PD((I - P)\theta + \theta'). \tag{40}$$

There are two cases: (1) $PD((I - P)\theta + \theta') = 0$ and (2) $PD((I - P)\theta + \theta') \neq 0$.

Like before, there are two cases. For case (2), the proof is identical, and so we omit it. For case (1), it must be the case that for some $P$, and $\theta'$

$$\ell((I - 2P)\theta + \theta' + \theta_0) = \ell(\theta + \theta' + \theta_0). \tag{41}$$

This is possible if and only if $P(\theta' + \theta_0) = \theta_i^\dagger$ for some $i$ and $P = P_i$ for the corresponding projection matrix. However, because $P\theta' = 0$, this requires that

$$P(\theta' + \theta_0) = \theta_i^\dagger. \tag{42}$$

By the definition of the reflection symmetry, we have that

$$P\theta' = \theta'. \tag{43}$$

This means that

$$\theta' = \theta_i^\dagger - P\theta_0. \tag{44}$$

At the same time, we have

$$PD((I - P)\theta + \theta') = 0, \tag{45}$$

which implies that

$$PD(I - P)\theta = -PD\theta'. \tag{46}$$

Because the right hand side is a constant that only depends on $\theta$. This can only happen if both sides are zero, which is achieved if:

$$PD(I - P) = 0, \tag{47}$$

and

$$\theta_i^\dagger = P\theta_0. \tag{48}$$

The first condition implies that

$$PD = PDP = D^T P^T = DP, \tag{49}$$

which implies that $P$ and $D$ must share the eigenvectors because they commute. Noting that

$$P\theta_i^\dagger = \theta_i^\dagger, \tag{50}$$

we obtain that $\theta_i^\dagger$ is an eigenvector of $P$ and so $\theta_i^\dagger$ is an eigenvector of $D$, but $D$ is diagonal and with nonidentical diagonal entries, $\theta_i^\dagger$ much then be a one-hot vector, and $P$ must also be diagonal and consists of values of 1 and 0 in the diagonal entries.

$\quad\square$

## A.7 PROOF OF THEOREM 3

*Proof.* First of all, note that shifting parameters of $\ell$ is the same as shifting the parameters of the weight decay. Therefore, for any local minimum $\theta'$,

$$\ell_{\mathrm{ar}}(\theta') = \ell(\theta) + \gamma\|\theta - \theta_0\|^2, \tag{51}$$

where $\theta = \theta' + \theta_0$.

Therefore,

$$\ell_{\mathrm{ar}}(\theta' + \theta^*) - \ell_{\mathrm{ar}}((I - 2P)\theta' + \theta^*) \tag{52}$$

$$= \ell(\theta + \theta^*) + \gamma\|\theta + \theta^* - \theta_0\|_D^2 - \ell((I - 2P)\theta + \theta^*) - \gamma\|(I - 2P)\theta + \theta^* - \theta_0\|_D^2 \tag{53}$$

$$= \gamma\|\theta + \theta^* - \theta_0\|_D^2 - \gamma\|(I - 2P)\theta + \theta^* - \theta_0\|_D^2 \tag{54}$$

$$= \gamma(z_\perp^T(D - I)z_\| + z_\perp^T D(\theta^* - \theta_0)), \tag{55}$$

where we have used the definition of reflection symmetry in the third line. In the fourth line, we have defined $z_\perp = P\theta$ and $z_\| = (I - P)\theta$. Thus,

$$\|\theta\|^2 = \|z_\perp\|^2 + \|z_\|\|^2 = \Theta(1). \tag{56}$$

Thus, we have that

$$\Theta(\ell_{\mathrm{ar}}(\theta' + \theta^*) - \ell_{\mathrm{ar}}((I - 2P)\theta' + \theta^*)) = \gamma\Theta(z_\perp^T(D - I)z_\|) + \Theta(z_\perp^T D(\theta^* - \theta_0)) \tag{57}$$

$$= \gamma\Theta(\sigma_D\|z_\|\|) + \Omega(\sigma_0\|z_\perp\|), \tag{58}$$

where we have used the fact that each element of $\theta^* - \theta_0$ is $\Omega(\sigma_0)$ because $\theta^*$ is an arbitrary constant and $\theta_0 \sim \mathcal{N}(0, \sigma^2)$. By the assumption that $\sigma_D = o(\sigma_0)$, we obtain the desired relation

$$\ell_{\mathrm{ar}}(\theta' + \theta^*) - \ell_{\mathrm{ar}}((I - 2P)\theta' + \theta^*) = \Omega(\gamma\sigma_0)\|z_\perp\|. \tag{59}$$

This finishes the proof. $\qquad\square$

## A.8 PROOF OF THEOREM 4

*Proof.* First of all,

$$\Pr(\min_i|\Delta_i| > \epsilon) = \Pr(|\Delta_1| > \epsilon \wedge ... \wedge |\Delta_N| > \epsilon) \tag{60}$$

$$\geq \max(0, \sum_i^N \Pr(|\Delta_i| > \epsilon) - N + 1), \tag{61}$$

where we have applied the Frechet inequality in the second line.

The sum $\sum_i^N \Pr(|\Delta_i| > \epsilon)$ can also be lower bounded if each $\Delta_i$ is a Gaussian variable with variance $\sigma_i$:

$$\sum_i^N \Pr(|\Delta_i| > \epsilon) \geq \sum_i^N(1 - \frac{2\epsilon}{\sqrt{2\pi\sigma_i^2}}) \tag{62}$$

$$\geq N - \frac{2\epsilon N}{\min_i\sqrt{2\pi\sigma_i^2}}. \tag{63}$$

Thus, for $\Pr(\min_i|\Delta_i| > \epsilon)$ to be larger than $1 - \delta$, we must have

$$\min_i\sigma_i \geq \frac{2\epsilon N}{\sqrt{2\pi}(1 - \delta)}. \tag{64}$$

Now, we show that $\Delta_i$-s are indeed Gaussian variables. From the previous proof, it is easy to see that for a unit vector $n_i$,

$$\Delta_i = \gamma n_i^T(\theta^* - \theta_0) + o(\gamma\sigma_0). \tag{65}$$

Therefore, $\Delta_i$ is a Gaussian variable with standard deviation $\gamma\sigma_0$. Thus, $\min_i\sigma_i = \gamma\sigma_0$. Thus, we have obtained the desired result

$$\gamma\sigma_0 \geq \frac{2\epsilon N}{\sqrt{2\pi}(1 - \delta)}. \tag{66}$$

$\qquad\square$

## A.9    PROOF OF THEOREM 5

Before we start proving Theorem 5, we introduce a definition that facilitates the proof.

**Definition 2.** *(Symmetry reduction.) We say that removing a symmetry from group $G_1$ reduces to removing the symmetry due to group $G_2$ if for any vector $n$,*

$$\|(I - \overline{G_2})n\| \le \|(I - \overline{G_1})n\|. \tag{67}$$

Now, we are ready to prove the main theorem.

*Proof.* We first show $(I - \overline{V})^T \nabla_\theta \ell(\theta) = 0$. For any $g \in V$ and $z \in \mathbb{R}$, we have

$$\ell(\theta + zn) = \ell(g(\theta + zn)), \tag{68}$$

where $n$ is an arbitrary unit vector. Taking the derivative with respect to $z$, and recalling that $g\theta = \theta$, we have

$$(gn)^T \nabla_\theta \ell(\theta) = n^T \nabla_\theta \ell(\theta). \tag{69}$$

Accordingly, we have

$$(Vn)^T \nabla_\theta \ell(\theta) := \frac{1}{|V|} \sum_{g \in V} (gn)^T \nabla_\theta \ell(\theta) = n^T \nabla_\theta \ell(\theta). \tag{70}$$

Due to the arbitrary choice of $n$, we have $(I - \overline{V})^T \nabla_\theta \ell(\theta) = 0$.

Therefore,

$$(I - \overline{V})^T \nabla_\theta \ell_{\mathrm{ar}}(\theta) = \gamma (I - \overline{V})^T \nabla_\theta \|\theta - \theta_0\|_D^2 \tag{71}$$

$$= 2\gamma (I - \overline{V})^T D(\theta - \theta_0) \tag{72}$$

$$= 2\gamma (I - \overline{V})^T \theta_0 + o(\gamma \sigma_0 (1 + \sigma_D)). \tag{73}$$

As $\theta_0$ is a Gaussian vector with mean $0$ and variance $\sigma_0^2$, $\|(I - \overline{V})^T \theta_0\|$ is a Gaussian variable with mean $0$ and variance $\|I - \overline{V}\|^2 \sigma_0^2 = \Omega(\mathrm{rank}(I - \overline{V})\sigma_0^2)$, which gives $\|(I - \overline{V})\nabla_\theta \ell\| = \Omega(\gamma \sigma_0 \sqrt{\mathrm{rank}(I - \overline{V})})$.

Now, we prove part (2) of the theorem. Note that if $V \lhd U$ and if $\theta_0 \in \ker \overline{V}$, then $\theta_0 \in \ker \overline{U}$. This means that for any group $U$ such that $V \lhd U$ and any vector $\theta_0$

$$\|(I - \overline{V})\theta_0\| < \|(I - \overline{U})\theta_0\|. \tag{74}$$

This means that to remove the symmetry from a large group $U$, it suffices to remove the symmetry from one of its minimal subgroups. Thus, let $M_G$ denote the set of minimal subgroups of the group $G$, we have

$$\min_{V \lhd G} \|(I - \overline{V})\theta_0\| \ge \min_{V \lhd M_G} \|(I - \overline{V})\theta_0\|. \tag{75}$$

The number of minimal subgroups is strictly upper bounded by the number of elements of the group because all minimal subgroups are only trivially intersect each other. This follows from the fact that the intersection of groups must be a subgroup, which can only be the identity for two different minimal subgroups. Therefore, the number of minimal subgroups cannot exceed the number of elements of the group. This finishes the second part of the theorem.

For the third part, we show that the symmetry broken subspace of any subgroup contains the symmetry broken subspace of a group generated by one of the generators and so it suffices to only remove the symmetries due to the subgroups generated by each generator. Let us introduce the following notation for a matrix representation $z$ of a group element:

$$\overline{z} = \frac{1}{\mathrm{ord}(z)} \sum_{i=1}^{\mathrm{ord}(z)} z^i, \tag{76}$$

where $\mathrm{ord}(z)$ denotes the order of $z$. This is equivalent to the symmetry projection matrix of the subgroup generated by $z$.

Now, let $G$ be abelian. Then, both $U$ and $V$ are abelian. Let us denote by $\Gamma(U) = \{z_i\}$ the mininal generating set of $U$. Suppose that for all $n \neq 0$ such that $n \in \text{im}(I - \overline{V})$, we must have $n \notin \text{im}(I - \overline{z_j})$ for all $j$. This means that

$$n \notin \bigcup_j \text{im}(I - \overline{z_j}). \tag{77}$$

Or, equivalently,

$$n \in \bigcap_j \text{im}(\overline{z_j}). \tag{78}$$

However, the space $\bigcap_j \text{im}(\overline{z_j}) \subseteq \text{im}(\overline{V})$ because $V$ is a subgroup of $U$, which is generated by $z_1, \cdots, z_m$. To see this, let $n \in \text{im}(\overline{z_j})$ for all $j$, then,

$$z_j n = n \tag{79}$$

for all $j$. Now, let $v = \prod_i z_i^{d_i(v)} \in V$, we have

$$(I - \overline{V})n = (I - \sum_v \prod_i z_i^{d_i(v)})n = 0 \tag{80}$$

This means that $n$ is in both $\text{im}(\overline{V})$ and $\text{im}(I - \overline{V})$, which is possible only if $n = 0$ – a contradiction. Therefore, as long as $I - \overline{V}$ is not rank 0, it must share a common subspace with one of the $I - \overline{z_j}$, and so removing the symmetry from any subgroup $V$ of $U$ can be reduced to removing the symmetry from the cyclic group generated by one of its generators from the minimal generating set.[8]

Therefore, we have proved that removing symmetries due to any subgroup of $U$ can be reduced to removing the symmetry from a (proper or trivial) subgroup of each of the cyclic decompositions of the group $U$, each of which is generated by a minimal generator of $U$. By the fundamental theorem of finite abelian groups, each of these groups is of order $p^k$ for some prime number $p$. Because each of these groups is cyclic, it contains exactly $k$ nontrivial subgroups. Taken together, this means that if $|U| = p_1^{k_1}...p_m^{k_m}$, we only have to remove symmetries from at most

$$\sum_i^m k_i = \log|U| \tag{81}$$

many subgroups. This completes part (3).

For part (4), we denote $\Delta_i := (I - \overline{V_i})^T \nabla_\theta \ell_{\text{ar}}(\theta)$ for $i = 1, \cdots, |\Gamma(U)|$. According to (73), $\Delta_i$ is approximately a Gaussian variable with zero mean and variance $\gamma^2 \text{rank}(I - \overline{V})\sigma_0^2$. Therefore,

$$\sum_i^{|\Gamma(U)|} \Pr(|\Delta_i| > \epsilon) \geq |\Gamma(U)| - \frac{2\epsilon|\Gamma(U)|}{\min_i \sqrt{2\pi\gamma^2\text{rank}(I - \overline{V})\sigma_0^2}}. \tag{82}$$

For $\Pr(\min_i |\Delta_i| > \epsilon)$ to be larger than $1 - \delta$, we must have

$$\gamma\sigma_0 \geq \frac{2\epsilon|\Gamma(U)|}{\sqrt{2\pi \text{ rank}(I - \overline{V})(1 - \delta)}}. \tag{83}$$

$\square$

---

[8]This holds true even if $V$ is a subset of $\langle z_j \rangle$.

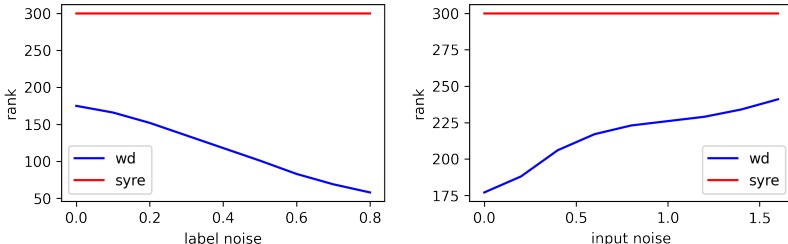

Figure 7: For the same setting as Figure 2, the rank of the covariance matrix decreases with the label noise and increases with the input noise. We set eigenvalues smaller than $10^{-3}$ to zero.

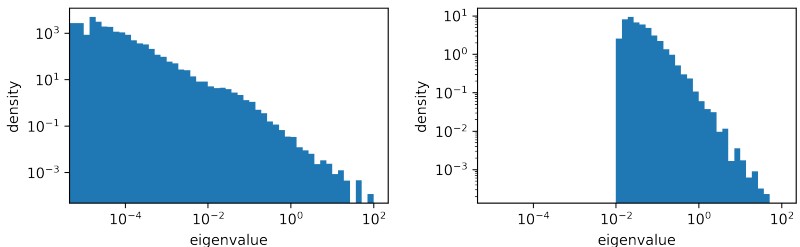

Figure 8: The spectrum of the covariance matrix of the model with vanilla weight decay (**Left**) and the *syre* model (**Right**) for $\gamma = 0.01$ and $\alpha = 1$. Clearly, the vanilla model learns a low-rank solution.

## B  ADDITIONAL EXPERIMENTS AND EXPERIMENTAL DETAIL

### B.1  TEACHER-STUDENT SCENARIO

This section gives some additional details and additional experiments in the teacher-student scenario in Figure 2. Specifically, we implement a two-layer network with tanh activation, 300 hidden units, and different input units. The network outputs a ten-dimensional vector corresponding to ten different classes. We then randomly generate such a network as the teacher model, 10000 standard Gaussian samples as the training set, and 1000 standard Gaussian samples. For both the *syre* and the vanilla model, we choose the Adam optimizer, learning rate 0.01, and weight decay 0.01.

As additional experiments, we also measure the influence of noisy labels and noisy input on the rank of the model in Figure 7. For the label noise, we randomly change 0% to 80% of the labels, and for the input noise, we add a Gaussian noise to the input with standard deviation 0 to 1.6. Figure 7 suggests that the rank of the vanilla model decreases in the face of noisy labels and increases in the face of noisy input, perhaps because the latter can be regarded as data augmentation. The *syre* model, however, is not affected.

### B.2  SUPERVISED LEARNING

This section presents some additional experiments for Section 6.3. Figure 8 gives the eigenvalue distribution of the networks in Fig.5, which further supports the claim that the vanilla network leads to a low-rank solution. In all the experiments above and in Section 6.3, we use a four-layer FCN with 300 neurons in each layer trained on the MNIST dataset with batch size 64.

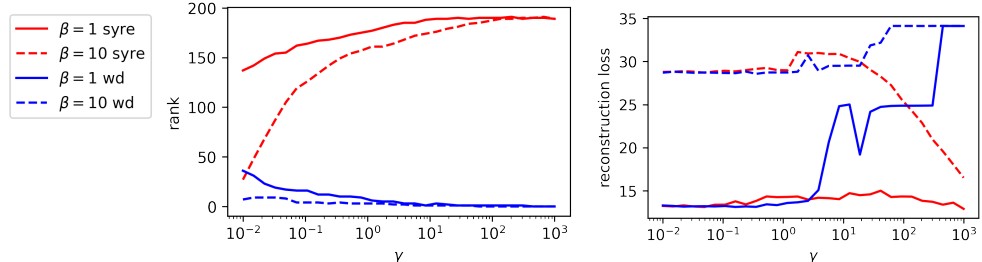

Figure 9: Rank and reconstruction loss for a VAE is trained on the Fashion MNIST dataset. The covariance of the model with vanilla weight decay has a low-rank structure and larger reconstruction loss. More importantly, posterior collapse happens at $\beta = 5$ but is mitigated with weight decay. **Left**: the rank of the encoder output with batch size 1000. We set eigenvalues smaller than $10^{-6}$ to 0. **Right**: reconstruction loss of vanilla and *syre* models.

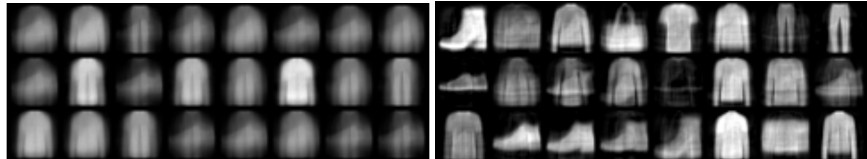

Figure 10: Examples of Fashion MNIST reconstruction with *syre* and $\beta = 10$. **Left**: No weight decay. **Right**: $\gamma = 1000$. Clearly, the posterior collapse is mitigated by imposing *syre* with weight decay.

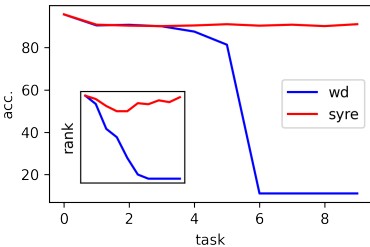

Figure 11: Performance and accuracy of a CNN trained on a continual learning task (permuted MNIST (Goodfellow et al., 2013; Kirkpatrick et al., 2017)). The main figure shows the test accuracy, and the subfigure shows the rank of the convolution layers output with batch size 1000, where we set eigenvalues smaller than $10^{-4}$ to 0. The results suggest that the rank of the model with vanilla weight decay gradually decreases, and the model completely collapses after the sixth task, while the *syre* model remains unaffected.

## B.3    POSTERIOR COLLAPSE

See Figures 9 and 10.

## B.4    CONTINUAL LEARNING

See Figure 11.

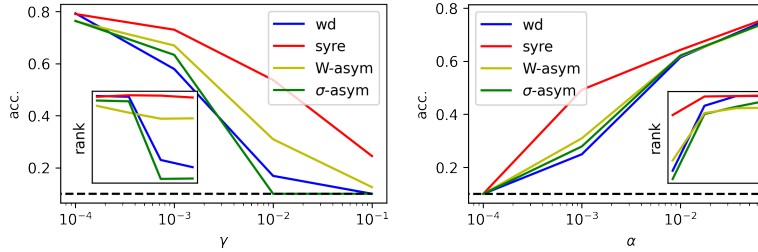

Figure 12: Test accuracy and rank of ViT for CIFAR10 (adapted from Yoshioka (2024)) with various weight decay $\gamma$ and data distributions with varyings strengths of linear correlation $\alpha$. As the theory predicts, the covariance of features of the model with vanilla weight decay has a low-rank structure and performs significantly worse. Dashed black lines correspond to random guesses. Subfigures show the rank of the covariance matrix of the final layer input with batch size 512. We set eigenvalues smaller than $10^{-4}$ to 0. **Left**: $\alpha = 1$ and different $\gamma$. **Right**: $\gamma = 0.0001$ and different $\alpha$.

| batchsize | MLP | *syre*-MLP | ResNet | *syre*-ResNet | ViT | *syre*-ViT |
|---|---|---|---|---|---|---|
| 512 | $95 \pm 0.1$ | $97 \pm 1$ | $18.4 \pm 0.7$ | $19.2 \pm 0.9$ | $80 \pm 1$ | $84 \pm 2$ |
| 256 | $48 \pm 1$ | $49 \pm 1$ | $18.6 \pm 0.2$ | $19.6 \pm 0.4$ | $48 \pm 5$ | $57 \pm 3$ |
| 128 | $26 \pm 1$ | $26 \pm 1$ | $19.9 \pm 0.4$ | $20.9 \pm 0.3$ | $39 \pm 1$ | $47 \pm 2$ |

Table 2: The per-batch time (ms) of various models.

| batchsize | MLP | *syre*-MLP | ResNet | *syre*-ResNet | ViT | *syre*-ViT |
|---|---|---|---|---|---|---|
| 512 | 382 | 384 | 7020 | 7049 | 4216 | 4218 |
| 256 | 360 | 382 | 3936 | 4254 | 2482 | 2484 |
| 128 | 360 | 382 | 2096 | 2190 | 1574 | 1574 |

Table 3: The GPU memory usage (M) of various models.

## B.5 VISION TRANSFORMER (VIT)

Similar to Section 6.3, we train the vanilla and *syre* ViT with various levels of weight decay $\gamma$ and various levels of input-output covariance $\alpha$. The dataset is constructed by rescaling the input by a factor of $\alpha$ for the CIFAR10 dataset. As the theoretical prediction, we can see that *syre* removes the rotation symmetries symmetry of transformers (Kobayashi et al., 2024) and leads to a full-rank solution with higher accuracy, as shown in Figure 12. Moreover, *syre* also performs better than the W-asymmetric and $\sigma$-asymmetric networks proposed in Lim et al. (2024). We implement $\sigma$-asymmetric networks using the recommended hyperparameters in Lim et al. (2024), and implement W-asymmetric networks by setting the same hyperparameters for each layer and doing a grid search.

## B.6 TIME AND MEMORY CONSUMPTION

The time and memory consumption of various models used in previous sections is given in Tables 2 and 3. In all experiments, we train the models on the CIFAR10 dataset with a signal A6000 GPU. It is clear that *syre* has a neglectable influence on time and memory consumption. This is as expected because *syre* only adds a static bias to training.

## B.7 LINEAR MODEL CONNECTIVITY

We also test the influence of *syre* on the linear model connectivity with the same setting as Lim et al. (2024). Specifically, we obtain two well-behaved MLP with parameters $\theta_1$ and $\theta_2$, and test the performance of another MLP with parameters $\lambda\theta_1 + (1 - \lambda)\theta_2$ for $0 < \lambda < 1$. By removing the permutation symmetry of MLP, we expect that the MLP with parameters $\lambda\theta_1 + (1 - \lambda)\theta_2$ also performs well. Figure 13 suggests that *syre* performs similarly to W-asymmetric networks proposed in Lim et al. (2024), while our methods have much fewer hyperparameters.

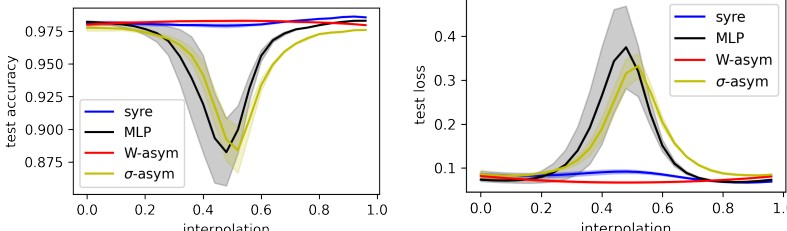

Figure 13: Linear mode connectivity: test accuracy and loss curves along linear interpolations between trained networks. We train MLPs over the MNIST dataset.

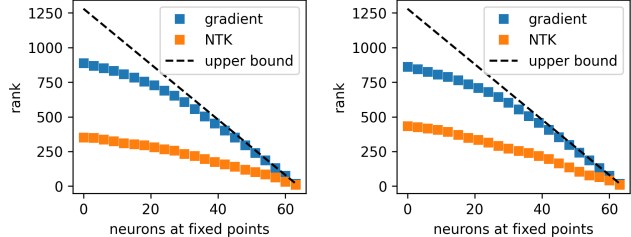

Figure 14: Ranks of the gradient covariance and NTK (**Left**: beginning of training. **Right**: end of training.) against the theoretical upper bound implied by Propositions 3 and 2. Two observations agree with our expectation: (1) the ranks of both matrices decrease as more and more neurons are stuck at the entrapped subspace, and (2) the ranks are always upper bound by the theoretical upper bound.

## C  SYMMETRY AND RANK OF THE NTK

In this section, we show that the rank of the NTK and gradient reduces as more and more of the neurons are in the symmetric state (namely, the fixed point induced by the group averaging projector).

Due to the difficulty in computing the spectrum of the NTK, we restrict to a two-layer subnetwork within a large network trained on MNIST with the number of neurons $10 \rightarrow 64 \rightarrow 10$, with 1280 many parameters. This means that the gradient second moment has dimension $1280 \times 1280$, and the model is trained on 1500 data points. Thus, the empirical NTK can be seen as a matrix having dimension $1500 \times 1500$. Here, the symmetry projection we consider is due to the permutation symmetry, whose projection has a rank of 20 (because every neuron has ten outgoing weights and incoming weights). Thus, according to the Propositions 3 and 2, the maximum possible rank for the NTK and the gradient is $1280 - 20 \times n$, where $n$ is the number of neurons projected to the symmetry fixed point. See Figure 14.

