# OpenReview forum: "Remove Symmetries to Control Model Expressivity and Improve Optimization"
_ICLR.cc/2025/Conference — ICLR 2025 Poster_

### Official Review · Reviewer_akeV · 2024-11-04

**Soundness:** 3
**Presentation:** 3
**Contribution:** 3
**Rating:** 6
**Confidence:** 2

**Summary:**

This paper addresses the problem of low-capacity states, often referred to as "collapse," that arise when neural network loss functions contain inherent symmetries. The authors prove two mechanisms through which symmetries reduce model expressivity, leading to these collapsed states and propose a method called syre to eliminate such symmetries. The syre algorithm introduces static random biases alongside weight decay, providing a simple yet effective means to remove almost all symmetry-induced constraints without requiring knowledge of the specific symmetries. The method is claimed to be model-agnostic, applicable to a broad range of architectures, and effective in both theoretical and practical contexts. Empirical results support the effectiveness of the proposed method in diverse settings, including supervised learning, self-supervised learning, and continual learning.

**Strengths:**

- The paper presents a novel approach to tackling symmetry-induced collapses in neural networks. The proposed method uses static random biases to remove symmetries, a simple yet innovative solution. The theoretical insights connect symmetry and expressivity in a new way, offering a significant advancement.
- The theoretical contributions are rigorous and well-supported with proofs. The authors also provide a range of empirical validations that align with their theoretical claims, making a strong case for the method’s effectiveness across various scenarios.
- The paper is clearly structured, with a logical flow from problem setup to solution. The core concepts are well-explained.
- The method's simplicity and general applicability make it impactful. By addressing a fundamental problem in deep learning training, syre has potential benefits for diverse architectures and tasks, opening new research avenues into model expressivity and optimization.

**Weaknesses:**

- Experiments are limited to small- or medium-scale models. The method’s applicability to real-world, large-scale tasks is insufficiently demonstrated. Benchmarks on high-impact datasets, such as ImageNet, and large-scale architectures, like transformers, would add credibility.
- The impact of completely removing symmetries on generalization is not well-discussed. Symmetry sometimes acts as a regularizer, and the absence of analysis on this potential downside is a gap.

**Questions:**

- How does syre scale to large models like transformers? Have you tested its efficiency or computational impact on such architectures?
- Would an adaptive bias approach improve the method compared to using a static bias throughout training?
- How sensitive is syre to hyperparameter choices, and do you have heuristics to guide practitioners in tuning them?

---

> ### Author Response · Authors · 2024-11-22
> **Rebuttal**
>
> Thank you for the detailed feedback. We will address both the weaknesses and the questions below.
>
> **Experiments are limited to small- or medium-scale models. The method’s applicability to real-world, large-scale tasks is insufficiently demonstrated. Benchmarks on high-impact datasets, such as ImageNet, and large-scale architectures, like transformers, would add credibility.**
>
> Thanks for this criticism. First of all, we would like to point out that some of our experiments are quite large-scale. For example, each reinforcement learning experiment takes roughly a week to run, longer than training CNNs on imagenet (often 1-2 days on our machines). The scale and difficulty of this experiment might be difficult to perceive unless familiar with RL. We also included a new experiment that implements syre on the vision transformer trained on the CIFAR10 dataset. Figure 12 in the revision shows that our methods work as expected on transformers, avoiding low-rank structures and achieving higher accuracy.
>
>
>
> **The impact of completely removing symmetries on generalization is not well-discussed. Symmetry sometimes acts as a regularizer, and the absence of analysis on this potential downside is a gap.**
>
> Thank you for your thoughtful question. Indeed, symmetries can sometimes act as implicit regularizers that benefit generalization (this point has been discussed in https://arxiv.org/abs/2309.16932, e.g., a point we have acknowledged in the conclusion), and the cases where having symmetry helps, of course, exist. The point we are making is that there are also cases where removing symmetry can be necessary: in our experiments (see Figures 3, 5, 6, and 11), we provide clear evidence of cases where our approach (syre) consistently achieves better generalization performance compared to the standard weight decay baseline ("vanilla" models).
>
> This improvement can be attributed to the fact that in our experimental settings, overfitting is not a significant concern. Instead, removing certain symmetries increases the model's expressiveness by enabling it to capture richer patterns in the data. By doing so, syre enhances both training and testing performance. We agree that in scenarios where overfitting is more pronounced, symmetry-induced regularization might indeed be beneficial. However, in the cases we analyzed, increasing the representation power by breaking these symmetries leads to superior generalization without causing overfitting. A deeper exploration of this trade-off in different regimes is an interesting direction for future work.
>
>
> **How does syre scale to large models like transformers? Have you tested its efficiency or computational impact on such architectures?**
>
> Thanks for this question. We have implemented syre on vision transformers in Section B5 of the revision, which shows that syre works well for transformers. Moreover, we test the memory and time consumption in Section B6 of the revision, verifying that syre has a neglectable effect on the training efficiency of various models (with an overhead of less than 5% and gets smaller as the training batch size gets larger).
>
>
> **Would an adaptive bias approach improve the method compared to using a static bias throughout training?**
>
> This is a good idea and might be true. There is a caveat to this; however, making the biases learnable also has the potential to introduce more symmetries to the loss function, and these symmetries may, in turn, reduce the effectiveness of the method. Therefore, making the biases adaptive needs to be done carefully, and this could be an interesting future problem.
>
>
>
>
>
> **How sensitive is syre to hyperparameter choices, and do you have heuristics to guide practitioners in tuning them?**
>
> This is another good question. We find the method to be quite robust and not so sensitive to the choice of hyperparameters. Since we have not found scenarios whether using a random weight decay $D$ is required, we mainly focused on studying the sensitivity of parameter $\sigma_0$.
>
> We discussed how the parameters could be chosen in Section 5.4. Often, we find that even a small bias (usually one or two orders of magnitudes smaller than the initialization scale) is sufficient to remove the negative effects of symmetry, while having an almost negligible effect on the performance of the model. This is what Figure-3 middle shows for Resnet18. A similar result is shown in the contrastive learning, where $\sigma_0=0.1$ or $0.01$ both improves the performance significantly and do not differ much from each other. However, using a very large $\sigma_0$ certainly hurts performance as it biases the parameters towards a Gaussian distribution.

---

> > ### Author Response · Authors · 2024-11-26
> >
> > Hi! We noticed that you have not communicated with us during the discussion period. It would be really helpful to us if we could hear more from you. We would be happy to know any additional thoughts or feedback you have on our work and on our previous reply!

---

> > > ### Comment · Reviewer_akeV · 2024-11-27
> > >
> > > Thank you for your detailed reply, which resolves my concerns. I will maintain my positive rating.

---

> ### Comment · Area_Chair_Uzmf · 2024-11-27
> **Response to Rebuttal**
>
> Do you mind letting the authors know if they have addressed your concerns and questions? Thanks!

---

### Official Review · Reviewer_PYEZ · 2024-11-05

**Soundness:** 3
**Presentation:** 3
**Contribution:** 2
**Rating:** 6
**Confidence:** 3

**Summary:**

This paper addresses the problem of symmetries in neural network loss landscapes and their role in causing models to get trapped in low-capacity states. The authors first prove that symmetries in the loss function can lead to reduced model capacity through two mechanisms: feature masking and parameter dimension reduction. They then propose SYRE, a simple modification to neural network training that aims to remove symmetries by adding a random bias to the parameters along with weight decay. For cases with uncountably many symmetries, they propose an advanced version that additionally uses a random diagonal matrix in the weight decay term. The method is model-agnostic and requires no knowledge of the underlying symmetries. The authors demonstrate their method's effectiveness across several applications where symmetry-induced capacity reduction is a concern, including supervised learning, variational autoencoders, self-supervised learning, and continual learning. They particularly show improvements in scenarios where models typically suffer from various forms of collapse, such as posterior collapse in VAEs and loss of plasticity in continual learning.

**Strengths:**

Here are the main strengths of the paper:

1. Addresses an important practical problem of symmetry-induced model collapse that affects multiple domains in deep learning (VAEs, continual learning, self-supervised learning).

2. Proposes a remarkably simple solution that requires minimal code changes and no architectural modifications, making it easily adoptable in practice.

3. Shows promising results in continual learning scenarios, demonstrating improved plasticity and performance maintenance over time compared to standard approaches.

4. Makes an interesting connection between symmetries and model collapse, providing a framework for thinking about how symmetries might limit model expressivity.

**Weaknesses:**

1. **Conceptual confusion about symmetries**:
   - The paper discusses "removing symmetries" but primarily analyzes fixed points where $P\theta_0 = 0$
   - This misses the key aspect of symmetries - their group orbits and the associated degeneracies
   - In the simple case of a two-layer linear network $F(x) = UVx$, the important symmetry structure lies in the $GL(h)$ orbits $(U,V) \to (Ug, g^{-1}V)$, not in fixed points

2. **Overly restrictive theoretical framework**:
   - The condition $P\theta' = \theta'$ for $(\theta', P)$-reflection symmetry is artificially restrictive
   - Simple examples show valid reflections that violate this condition
   - For instance, in 2D with $\theta' = (0,1)$ and reflection in x-direction $P = ((1,0),(0,0))$, we have $P\theta' \neq \theta'$ yet this is a perfectly valid reflection symmetry

3. **Questionable arguments about gradient vanishing**:
   - The paper suggests symmetric points trap models due to vanishing gradients
   - However, in the UV example, fixed points along $U=V$ line have nonzero gradients except at origin and minima
   - Gradients naturally vanish along symmetry orbits (they're level sets of the loss), but this doesn't imply trapping as claimed

4. **Limited analysis of regularization effects**:
   - The D-norm modification merely changes spherical weight decay to ellipsoidal weight decay
   - The $\theta_0$ shift changes the center of the regularization
   - While these modifications may improve optimization, it's not clear they "remove symmetries" as claimed
   - In the UV example, these modifications may select preferred points along symmetry orbits (like weight decay does) but may not fundamentally break the symmetry structure

5. **Experimental limitations**:
   - Lack of systematic comparison with standard weight decay baseline
   - Only a few experiments (particularly continual learning) show clear improvements over weight decay
   - Results suggest their method might be a useful variant of weight decay, but not necessarily for the theoretical reasons provided

The paper's suggested regularization may indeed have advantages over standard weight decay in certain scenarios, but the theoretical arguments about symmetry breaking are not convincing, and the empirical validation needs stronger baselines to support the claims.

**Questions:**

Those are good questions. Here are those formatted plus a few additional ones I think would be helpful:

1. **Fixed Points vs Symmetries**:
   - Do you mean fixed points (invariant sets) when discussing "symmetric points" where $P\theta_0 = 0$? If so, consider revising terminology to avoid confusion with symmetry orbits.
   - Why focus on fixed points rather than the full symmetry orbits which characterize the degeneracies in the loss landscape?

2. **Gradient Analysis**:
   - Can you clarify the argument around equation (2) about vanishing gradients? How does this differ from the natural vanishing of gradients along symmetry orbits?
   - In the expansion around fixed points, do you consider gradients in directions orthogonal to the symmetry?

3. **Initialization and Fixed Points**:
   - For a simple network $F(x) = UVx$, the origin $(U,V)=(0,0)$ is a fixed point, but why is this relevant when standard initialization (Glorot, etc.) avoids such points?
   - Do your arguments about trapping at fixed points apply to more realistic initialization schemes?

4. **Regularization Effects**:
   - Can you provide ablation studies comparing your method with standard weight decay?
   - For the D-norm modification, how does making weight decay ellipsoidal fundamentally differ from standard weight decay in terms of symmetry breaking?

5. **Theoretical Claims**:
   - The condition $P\theta' = \theta'$ seems unnecessarily restrictive. Can your results be generalized without this condition?
   - For continuous symmetries (like $GL(h)$ in linear networks), can you make precise what subset of symmetries your method breaks?

6. **Empirical Validation**:
   - Could you provide standard weight decay baselines for all experiments?
   - In scenarios where your method outperforms standard weight decay (like continual learning), what mechanism explains this improvement?

Would you like to add any other questions?

---

> ### Author Response · Authors · 2024-11-22
> **Rebuttal Part 1**
>
> Thank you for the detailed feedback. We will address both the weaknesses and the questions below.
>
> **Conceptual confusion about symmetries: The paper discusses "removing symmetries" but primarily analyzes fixed points where P\theta_0=0. This misses the key aspect of symmetries - their group orbits and the associated degeneracies.**
>
> Thank you for your question. We believe this criticism is both due to a misunderstanding of our theoretical framework and due to a lack of knowledge of the nature of the fixed points in the UV model (or the matrix factorization model).
>
> We point out that it is a foundational result in the theory of finite groups that any finite group has a fixed point (which can be constructed through the group average $\sum_g g$. E.g., see chapter 2 of: https://www.math.columbia.edu/~khovanov/finite2016/RepTheoryFinite2.pdf). This immediately implies that any group that contains a finite subgroup will also have these fixed points, and thus, removing these fixed points is the same as removing these groups. While the orbits and degeneracies are important, removing the fixed points is sufficient to remove these group structures.
>
> Lastly, it is not about "missing the group orbits" -- the orbits per se do not tend to create optimization entrapment (as you also point out below). It is the fixed points that do lead to these difficulties, and this is why removing fixed points is the focus of our work. See our answers below and references related to this point.
>
>
>
> **In the simple case of a two-layer linear network F(x)=UVx, the important symmetry structure lies in the GL(h) orbits, not in fixed points.**
>
> Unfortunately, we cannot agree that the fixed points are not important. For example, see https://arxiv.org/pdf/1612.09296, which shows that these fixed points are precisely the cause of the saddle points in the loss function, which needs to escape very carefully. From an optimization perspective, this is as important as (if not more important than) the orbits.
>
> Secondly, our focus on the $(\theta', P)$-reflection symmetry addresses a fundamental Z2 group symmetry (and any finite group structures, as Theorem 5 shows). Note that GL(h) symmetries must contain Z2 subgroup symmetries. We can thus remove the GL(h) symmetry by removing the symmetry of its Z2 subgroups. In the context of a two-layer linear network, we argue that breaking these GL(h) symmetries can be systematically approached by first removing the symmetries of its Z2 subgroups. For example, in linear networks with h hidden units, there exist h! permutation symmetries that can be removed together. Each permutation corresponds to a distinct fixed point, and by removing these, we must also remove the GL(h) structure.
>
> Moreover, although the linear network has infinitely many Z2 symmetries, we can effectively remove all of them by the method introduced in Section 5.2. More generally, Theorem 5 implies that we can remove any symmetry which contains a finite group symmetry.
>
> **Overly restrictive theoretical framework: The condition P\theta’=\theta’ for (\theta’,P)-reflection symmetry is artificially restrictive. Simple examples show valid reflections that violate this condition. For instance, in 2D with \theta’=(0,1) and reflection in x-direction  P=((1,0),(0,1)), we have P\theta’\neq\theta’, yet this is a perfectly valid reflection symmetry.**
>
> We are afraid to say that this is an understanding. The $(\theta’,P)$-definition is NOT a restriction but the most general way to define these fixed points/surfaces. To specify a mirror (or any surface), one needs to specify its (1) orientation and (2) location. The $P$ gives the orientation, and $\theta'$ gives the location of this surface.
>
> One can see that when $P$ is a surface, the choice of $\theta'$ is not unique. There exists $\theta'\neq \theta''$ such that $(\theta', P)$ and $(\theta'',P)$ refers to the same symmetry. This is exactly what Proposition 1 proves: If $P \theta' = P \theta''$, then $(\theta', P)$ and $(\theta'', P)$ are referring to the same symmetry! Thus, the requirement that $P\theta' =\theta'$ is not a restriction but is nothing but a notational simplification that is without loss of generality.
>
> To clarify with your example: the reflection you describe in the x-direction corresponds to what we define as a $(0,P)$-reflection, where $P = \begin{pmatrix} 1 & 0 \\ 0 & 1 \end{pmatrix}$. In this context, our framework would express the reflection as\theta \rightarrow (I - 2P) \theta for any vector \theta.

---

> ### Author Response · Authors · 2024-11-22
> **Rebuttal Part 2**
>
> **Questionable arguments about gradient vanishing: The paper suggests symmetric points trap models due to vanishing gradients. However, in the UV example, fixed points along U=V line have nonzero gradients except at origin and minima. Gradients naturally vanish along symmetry orbits (they're level sets of the loss), but this doesn't imply trapping as claimed.**
>
> We are afraid to say that this criticism is based on incorrect assumptions. The fixed points of the UV model are not just the origin and minima.
>
> Any low-rank solution is also a (or multiple) fixed point. If $U$ and $V$ are square matrices with dimension $d$, then for any $d'<d$, there is a low-rank saddle and escaping these low-rank solutions is impossible for gradient descent. This is due to the permutation symmetry (which is a discrete subgroup of GL(n)) of the UV model. This is a well-known fact. For example, see the reference we give above; also, see https://arxiv.org/pdf/2105.12221; also, see the discussion of rotation/permutation symmetry in https://arxiv.org/abs/2309.16932; in fact, this kind of results appear as early as in https://www.ism.ac.jp/~fukumizu/localmin3.pdf
>
> Usually, continuous groups per se do not cause entrapment. It is the discrete/finite subgroups of the continuous groups that actually cause these entrapments. In fact, the GL(n) group contains many discrete subgroups. Actually, this implies that if we removed any discrete subgroup of $GL(n)$, we must also have removed $GL(n)$ itself.
>
> **Limited analysis of regularization effects: The D-norm modification merely changes spherical weight decay to ellipsoidal weight decay. The \theta_0  shift changes the center of the regularization. While these modifications may improve optimization, it's not clear they "remove symmetries" as claimed. In the UV example, these modifications may select preferred points along symmetry orbits (like weight decay does) but may not fundamentally break the symmetry structure**
>
> This is a good question, but again, it is due to a misunderstanding of our theory. Our method can indeed remove the symmetries in this case. Also, according to our theory and previous works, it is not the orbits that hurt the optimization process but the fixed points introduced by the group projections. Also, as we explained above, your understanding of the role of \theta_0 and the D-norm is incorrect. As addressed in our previous responses, in the standard UV model with weight decay, solutions often converge to low-rank configurations due to the inherent symmetry structure. In contrast, our modifications (\theta_0θ shift and D-norm) alter the geometry of the optimization landscape, effectively preventing convergence to these low-rank, symmetric solutions.
>
> **Experimental limitations: Lack of systematic comparison with standard weight decay baseline. Only a few experiments (particularly continual learning) show clear improvements over weight decay. Results suggest their method might be a useful variant of weight decay, but not necessarily for the theoretical reasons provided.**
>
> Thank you for your valuable feedback. We believe there may be a misunderstanding regarding our comparisons with standard weight decay. In fact, we systematically compare our method (referred to as syre) with vanilla weight decay across Figures 1, 2, 3, 5, 6, 7, 8, 9, 10, and 11. In these figures, the term "vanilla" refers to models using the standard weight decay with the same \gamma. The results consistently demonstrate that syre outperforms vanilla weight decay across various settings, avoiding low-rank degeneracies and achieving superior performance.
>
> Moreover, as we have addressed in our previous responses, syre effectively breaks different kinds of symmetries. This mechanism prevents convergence to or entrapment at low-rank solutions, leading to the observed full-rank models, which contrasts clearly with the behavior under standard weight decay. Therefore, the improvements are not just incidental but align with our theoretical claims on symmetry removal, as demonstrated empirically in the figures.
>
>
> **Fixed Points vs Symmetries: Do you mean fixed points (invariant sets) when discussing "symmetric points" where P\theta_0=0? If so, consider revising terminology to avoid confusion with symmetry orbits. Why focus on fixed points rather than the full symmetry orbits which characterize the degeneracies in the loss landscape?**
>
> Thank you for this question. You are correct that what we refer to as "symmetric points" are indeed invariant sets where $P\theta_0 = 0$. We will stick to the terminology of "fixed point" to keep consistency.

---

> ### Author Response · Authors · 2024-11-22
> **Rebuttal Part 3**
>
> **Gradient Analysis: Can you clarify the argument around equation (2) about vanishing gradients? How does this differ from the natural vanishing of gradients along symmetry orbits? In the expansion around fixed points, do you consider gradients in directions orthogonal to the symmetry?**
>
> Thanks for this question. They are completely different, and this has been studied by many works. The vanishing gradient in Eq (2) implies the existence of saddle points. Again, see https://arxiv.org/pdf/1612.09296 and https://arxiv.org/abs/2309.16932 for good explanations. In contrast, the gradient vanishing along the orbits is not directly related to saddles (and is thus unrelated to entrapment, which you have been correct about).
>
>
> **Initialization and Fixed Points: For a simple network F(x)=UVx, the origin (0,0) is a fixed point, but why is this relevant when standard initialization (Glorot, etc.) avoids such points? Do your arguments about trapping at fixed points apply to more realistic initialization schemes?**
>
> Again, as we explained above, there are many many other fixed points, apart from the origin. If U and V are d by d matrices, then this model contains the $S_d$ finite subgroup, which is believed to have (super) exponentially many finite distinct subgroups. For example, if $d=10$, $S_n$ has $29594446$ many finite subgroups (https://groupprops.subwiki.org/wiki/Subgroup_structure_of_symmetric_groups)!
>
> There are many mechanisms for SGD or GD to be trapped in these solutions. On the one hand, these solutions tend to be attractive for SGD at a large learning rate through a mechanism called "stochastic collapse" (https://arxiv.org/abs/2306.04251). Another mechanism is that GD/SGD are attracted to these solutions when weight decay is used. See https://arxiv.org/abs/2309.16932. This is relevant to, for example, why transformers tend to learn low-rank solutions (see https://arxiv.org/abs/2410.23819). Therefore, one does not need to start close to these saddles to be trapped there.
>
>
>
> **Regularization Effects: Can you provide ablation studies comparing your method with standard weight decay? For the D-norm modification, how does making weight decay ellipsoidal fundamentally differ from standard weight decay in terms of symmetry breaking?**
>
> Thank you for your question. To clarify, in Figures 1, 2, 3, 5, 6, 7, 8, 9, 10, and 11, the term "vanilla" corresponds to models using standard weight decay with identical \gamma. We have systematically compared our method (syre) against these standard weight decay baselines in various experimental setups to show our improvement. In terms of the D-norm modification, it is only a theoretical guarantee that we can break infinite-many symmetries together, while empirically, we can effectively break all symmetries without the D-norm.
>
> **Theoretical Claims: The condition P\theta’=\theta’ seems unnecessarily restrictive. Can your results be generalized without this condition? For continuous symmetries (like GL(h) in linear networks), can you make precise what subset of symmetries your method breaks?**
>
> Thank you for this question. Please first see our answer to the related question above. Essentially, our method removes the symmetry of any group for which a fixed point projection exists -- which is guaranteed as long as this group contains a finite subgroup.
>
> **Empirical Validation:Could you provide standard weight decay baselines for all experiments? In scenarios where your method outperforms standard weight decay (like continual learning), what mechanism explains this improvement?**
>
> Thank you for your question. To clarify, in Figures 1, 2, 3, 5, 6, 7, 8, 9, 10, and 11, the term "vanilla" corresponds to models using standard weight decay with identical \gamma settings. We have systematically compared our method (syre) against these standard weight decay baselines in various experimental setups. The consistent improvement observed with syre arises from its ability to avoid convergence to low-rank solutions, which is a common limitation of standard weight decay. In particular, we empirically demonstrate that syre preserves higher ranks in the learned representations, leading to better performance across different tasks, including but not limited to continual learning.

---

> > ### Author Response · Authors · 2024-11-26
> >
> > Hi! We noticed that you have not communicated with us during the discussion period. It would be really helpful to us if we could hear more from you. We would be happy to hear any additional thoughts or feedback you have on our work and on our previous reply!

---

> > > ### Comment · Reviewer_PYEZ · 2024-11-26
> > > **score increase**
> > >
> > > Thank you for the thorough explanations. I now understand why you were focusing on the fixed points, and now the symmetry leading to saddle points makes sense. The low-rank argument also seems like a natural consequence of the 1D fixed points.
> > > I also appreciate clarifying that the vanilla experiments all had weight decay. Therefore, I will increase my score.

---

### Official Review · Reviewer_JSoA · 2024-11-07

**Soundness:** 3
**Presentation:** 2
**Contribution:** 2
**Rating:** 6
**Confidence:** 3

**Summary:**

Main contributions are:
- Builds on (Ziyin 2024) to introduce two ways that symmetry may reduce model capacity— a) that in the lazy training limit, the P-symmetry ‘masks’ certain features around a symmetric solution and b) the parameter dimension is reduced by the rank of P when around a symmetric solution.
- Introduces a straightforward approach to eliminate these symmetric solutions by both fixing a random bias on the neurons (to handle finite symmetries) and also modifying weight decay to use a norm with respect to a positive diagonal matrix D (to handle infinite symmetries, e.g. like those that appear in the residual stream of a transformer). Also introduce a measure to quantify the degree to which a symmetry is broken.
- Experiments that measure the extent of symmetry removal for some linear regression set-ups and some vision tasks, and continual learning in an RL setting.

**Strengths:**

- As noted, provide some plausible and interesting explanations for some mechanisms for how symmetry may reduce capacity that nontrivially build on (Zivin 2024).
- In some settings and experiments, their algorithm (syre) seems to usefully outperform baselines/another symmetry removal method?

**Weaknesses:**

Experimental shortcomings
- The authors make strong claims around the ability of symmetry-removal to improve performance for a) SSL, b) a supervised continual learning problem, and c) an RL continual learning problem. They do not implement proper baseline comparisons for any of these settings nor go beyond toy experiments. If they want to claim that (some degree of) symmetry removal is helpful for training models in practical settings, they need to either show that a) symmetry removal is reasonably competitive in mildly (compute/FLOP-controlled or at least aware) realistic settings or b) symmetry removal explains some % of the success of methods that do well in these settings. Aside from performance gains, benchmarking their method against the (Lim 2024) methods on linear mode connectivity would also help better validate their set-up. Likewise, benchmarking on other approaches on symmetry-removal.
- On the other hand, the authors only very loosely test their explanations in Prop. 2/3. Presumably, if this explanation is correct, shouldn’t models with more/less symmetries be less/more stuck in neuron-collapsed solutions (e.g. in the spirit of the Zivin 2024 experiments)? Fig. 1 only shows one case. If this claim around the science of models is to be taken as a main contribution of the paper, more experimental evidence of the propositions is needed.


Miscellaneous
- Typo on line 65-66: “$\|\theta\|_D^2:=w^T D w$ is the norm of $w$ with respect to $D$” should be “$|w|_D^2:=w^T D w$ is the norm of $w$ with respect to $D”


Ziyin, Liu. "Symmetry Induces Structure and Constraint of Learning." Forty-first International Conference on Machine Learning.
Lim, Derek, et al. "The Empirical Impact of Neural Parameter Symmetries, or Lack Thereof." arXiv preprint arXiv:2405.20231 (2024).

**Questions:**

- How does your syre do on the (Lim 2024) LMC experiments?
- Do you have any (even at a very high level) design rules for how a training-algorithm designer may usefully modulate the symmetries in a model?

---

> ### Author Response · Authors · 2024-11-22
> **Rebuttal**
>
> Thank you for the detailed feedback. We will address both the weaknesses and the questions below.
>
>
> **The authors make strong claims around the ability of symmetry-removal to improve performance for a) SSL, b) a supervised continual learning problem, and c) an RL continual learning problem. They do not implement proper baseline comparisons for any of these settings nor go beyond toy experiments.**
>
> We believe that these are misinterpretations of our claims. Please first see our explanation of this point in the summary rebuttal. Our actual claims are careful and substantially supported by experiments. The only major claim we have is that removing symmetries can avoid convergence to low-capacity states and nothing more.
>
> We do not claim that the proposed method achieves the SOTA performance of these methods. We only claim that the proposed method removes the symmetries in the loss function and thereby removes the low-capacity solutions -- these claims are justified by the experimental result that rank collapse no longer happens after applying our method. The point of these experiments is to show that the symmetries are indeed removed, as our theory claims.
>
> Also, as there is no method for agnostic symmetry removal at all, there are really no baselines to implement. The proper and primary comparison is the vanilla weight decay because our method can be interpreted as a variant of weight decay. Empirically, we demonstrate that these symmetry removals correlate with improved performance, which is validated with experiments.
>
>
> **If they want to claim that (some degree of) symmetry removal is helpful for training models in practical settings, they need to either show that a) symmetry removal is reasonably competitive in mildly (compute/FLOP-controlled or at least aware) realistic settings or b) symmetry removal explains some % of the success of methods that do well in these settings.**
>
> We apologize that we do not fully understand this question. On the compute-FLOP control side of the question, we have included a new experiment showing that the proposed method incurs essentially no overhead over the standard training method in MLP, resnets, and vision transformers. In terms of realistic settings, we have included a new experiment on vision transformers, which shows that the proposed method indeed makes the model much more robust to collapses. Note that the rank collapse problem in transformers has been identified to be due to the symmetries in the attention layer (see https://arxiv.org/abs/2309.16932 and, more recently, https://arxiv.org/abs/2410.23819). Again, the only major claim we have is that removing symmetries can avoid convergence to low-capacity states -- which have been fully experimentally and theoretically justified.
>
>
> **Aside from performance gains, benchmarking their method against the (Lim 2024) methods on linear mode connectivity would also help better validate their set-up. Likewise, benchmarking on other approaches on symmetry-removal.**
>
> We are afraid to say that we find this criticism unfair. There is really no method for agnostic symmetry removal, and our method is really the first of its kind.  While it is not quite reasonable or fair to compare it with Lim et al., we included a vision transformer training task that we also compared with the methods in Lim et al. in Appendix B.5, and the performance of our methods on linear mode connectivity in Appendix B.7. Our method performs competitively against the methods in Lim et al.

---

> ### Author Response · Authors · 2024-11-22
> **Rebuttal Part 2**
>
> **On the other hand, the authors only very loosely test their explanations in Prop. 2/3. Presumably, if this explanation is correct, shouldn’t models with more/less symmetries be less/more stuck in neuron-collapsed solutions (e.g. in the spirit of the Zivin 2024 experiments)? Fig. 1 only shows one case. If this claim around the science of models is to be taken as a main contribution of the paper, more experimental evidence of the propositions is needed.**
>
> Thanks for raising this question. We have added more experiments validating Propositions 3 and 2 (Section C). Here, we plot the rank of the gradient and NTK of a neural network before and after training. The results are shown to agree with the theoretical prediction of Propositions 2 and 3, where the ranks of these matrices decrease as more and more neurons are initialized at a symmetry-entrapped subspace. Also, we would like to point out that it is the case that having more symmetries leads to more entrapment, not less.
>
> **Typo on line 65-66: “ |\theta|_D^2=w^TDw is the norm of w with respect to D” should be “|w|_D^2=w^TDw is the norm of w with respect to D”.**
>
> Thank you for pointing this out. We have corrected this typo.
>
> **How does your syre do on the (Lim 2024) LMC experiments?**
>
> Thanks for suggesting this. While we think Lim et al. (2024) have achieved very interesting results in the LMC experiments, we point out that there is really no need nor fair for us to compare the two methods. See our explanation of this point in the summary rebuttal.
>
> The two papers really have different focuses. Our focus is on how to remove an agnostic symmetry. The focus of Lim et al. is on some of the consequences if one removes the permutation symmetry. There is also no need to test our method for the LMC experiments, as our algorithm is not designed for this purpose. The comparison will also not be fair for two reasons. First of all, our method is symmetry agnostic, whereas Lim's method is particularly designed to tackle permutation symmetries. It does not make much sense to compare a general method to a specific technique. This is essentially comparing apples to oranges.
>
> Also, it is very easy to construct cases that Lim et al. (2024) completely fails, while our proposed method can still run robustly. We presented one such example in Figure 4. Another rather extreme but illustrative example is a deep network with width 1. Here, Lim et al. (2024)'s method fails completely (as it simply prevents the whole model from training), whereas our method can still be applied and can work well in principle.
>
> That being said, we do provide one such comparison with the two baselines in Lim et al. (2024) on an MLP network. Our proposed method is shown to work much better than $\sigma$-asym, which also has a single hyperparameter to tune, like our method, and performs similarly to W-asym, which has $8$ hyperparameters to tune for this simple model.
>
>
> **Do you have any (even at a very high level) design rules for how a training-algorithm designer may usefully modulate the symmetries in a model?**
>
> Thank you for your question. We suggested a high-level design guideline in the conclusion section. Specifically, models with excessive symmetry may fall into low-capacity traps, while models with too little symmetry may be prone to overfitting. Thus, one possible design logic is to gradually reduce symmetries until the onset of overfitting.

---

> > ### Comment · Reviewer_JSoA · 2024-11-23
> > **Comment on the rebuttal**
> >
> > I appreciate the new experiments, especially those testing Prop. 2/3 and the new LMC experiments. Some additional points I would appreciate the authors addressing, if possible:
> >
> > - **Regarding claims of increased performance.** I think it would significantly clarify your contribution if you put in your paper, as you note in this rebuttal, that symmetry removal *correlates* with improved performance instead of making what might be interpreted as stronger or causal claims about the absolute performance. For example, from your abstract: "*The proposed method is shown to improve the training of neural networks in scenarios when this type of entrapment is especially a concern.*" and from section 6: "*We then apply the method to a few settings where symmetry is known to be a major problem in training. We see that the algorithm leads to improved model performance on these problems.*" Readers might be confused about either the prevalence of such entrapment (which seems to be an open question and therefore makes some of these results quite difficult to interpret) or the strength of the methods (that they outperform other more sophisticated regularization or algorithms, which might as a side effect also solve the entrapment problem). If the training performance over simple baselines are simply used as evidence for syre's effectiveness, it should be stated.
> >
> > - **On the LMC results.** I requested this experiment not as an evaluation for your method against Lim et al, but as additional evidence of symmetry removal (which, as an aside, I think provide far stronger evidence than your performance experiments!). I am quite happy that these new experiments are included. However, I am confused by the statement in your rebuttal that using syre in this setting is somehow "far beyond the case of linear mode connectivity problems." Could the authors explain?

---

> > > ### Author Response · Authors · 2024-11-23
> > > **Reply**
> > >
> > > Thanks for your additional comment.
> > >
> > > **Regarding claims of increased performance.** Thanks for this suggestion. We have updated the manuscript to emphasize that removing the symmetries only correlates with an improved performance. We will also carefully check through our manuscript again before the final version to avoid similar phrasings that might mislead the readers.
> > >
> > >
> > > **On the LMC results.** Thanks for your explanation, and we appreciate your suggestion, as it reveals an additional usage of our method that we previously did not think about.
> > >
> > > Also, we apologize for the unintended confusion. What we intended to say by "far beyond the case of linear mode connectivity problems" is that we think the problem of removing symmetries is a much bigger and more general problem --and perhaps a more fundamental problem. Our method is designed as a general-purpose method for removing symmetries, which may be of use to many other problems and is not just limited to improving the specific problem of LMC.

---

> > > > ### Comment · Reviewer_JSoA · 2024-11-25
> > > > **score increase**
> > > >
> > > > Thanks for making these changes. I have increased my score.

---

### Official Review · Reviewer_Fj68 · 2024-11-12

**Soundness:** 3
**Presentation:** 3
**Contribution:** 3
**Rating:** 6
**Confidence:** 3

**Summary:**

This paper presents a novel approach for addressing the limitations posed by symmetries in neural network training, which often lead to low-capacity model states. The authors propose a method called "syre", which can effectively remove almost all types of symmetries from neural networks, thereby enhancing model expressivity and performance across various training scenarios. The method is both model and symmetry group agnostic and straightforward to implement, requiring only minimal changes to existing training pipelines.

**Strengths:**

1. The proposed method is theoretically sound. Authors have rigorously proved many useful results like Proposition 3 which shows symmetry reduces parameter dimension.
2. Their proposed solution is generalizable across different architectures and does not require detailed knowledge of the model’s symmetries.
3. The empirical evaluations demonstrate significant improvements in model performance and capacity, particularly in challenging scenarios like continual learning and self-supervised learning setups.

**Weaknesses:**

1. Some of the theoretical explanations are dense and might be challenging for readers not familiar with the detailed aspects of symmetry in neural networks. Having decent knowledge of symmetries in neural networks I still struggled a bit with some sections like Section 5. Summarizing early on what the main theoretical result of each section might make it easier for reader to appreciate the contributions of the paper better.

2. The limitations of the method are not thoroughly discussed, which could be beneficial for practitioners to know when to use this technique with caution.

3. The formatting of the paper seems off. All the headings are pretty close to the previous paragraph hurting the readability.

**Questions:**

1. It would be beneficial to see further exploration into the limits of the method, especially in more complex or less controlled environments than those tested. Also, maybe trying some transformer architecture based methods for SSL might make the architecture agnostic point stronger?

2. Could the authors elaborate on any potential computational overhead or scalability issues when implementing syre in larger or more complex models?

---

> ### Author Response · Authors · 2024-11-22
> **Rebuttal**
>
> Thank you for the detailed feedback. We will address both the weaknesses and the questions below.
>
> **Some of the theoretical explanations are dense and might be challenging for readers not familiar with the detailed aspects of symmetry in neural networks. Having decent knowledge of symmetries in neural networks I still struggled a bit with some sections like Section 5. Summarizing early on what the main theoretical result of each section might make it easier for reader to appreciate the contributions of the paper better.**
>
> Thank you for your constructive feedback. In response, we have added brief summaries at the beginning of Sections 5.1, 5.2, and 5.3 to clarify the main theoretical results. For instance, at the start of Section 5.1, we now state: “This subsection will show that when the number of reflection symmetries in the loss function is finite, one can completely remove them using a simple technique,” which highlights the main implication of Theorem 1.
>
>
> **The limitations of the method are not thoroughly discussed, which could be beneficial for practitioners to know when to use this technique with caution.**
>
> Thank you for pointing this out. We discussed the limitations briefly in the conclusion section. Specifically, while our method efficiently removes symmetries, it does not always guarantee improved performance, as “a model completely without symmetry may have undesirably high capacity and be more prone to overfitting.” Thus, practitioners should selectively retain beneficial symmetries while removing undesired ones.
>
> Also, another limitation is demonstrated systematically by Figure 4: shifting the initialization creates a small error proportional to the amount of shifting, and thus, there is a tradeoff between how much shifting one wants to achieve and how much error in the objective function one can tolerate. There could be other limitations that we are not yet aware of, and these could be interesting directions for future research.
>
>
>
> **The formatting of the paper seems off. All the headings are pretty close to the previous paragraph hurting the readability.**
>
> Thanks for the criticism. We have fixed the spacing of the headings.
>
>
>
> **It would be beneficial to see further exploration into the limits of the method, especially in more complex or less controlled environments than those tested. Also, maybe trying some transformer architecture based methods for SSL might make the architecture agnostic point stronger?**
>
> Thanks for suggesting this experiment. We have now included an experiment of vision transformers trained on CIFAR10. Our code is adapted from github.com/kentaroy47/vision-transformers-cifar10. Figure 12 in the revision shows that our methods work as expected on transformers, avoiding low-rank structures and achieving higher accuracy.
>
>
> **Could the authors elaborate on any potential computational overhead or scalability issues when implementing syre in larger or more complex models?**
>
> Thanks for this question. Theoretically, the method has the same computational complexity as the standard training methods, and so there is no difference in principle.
>
> Practically and empirically, we notice almost zero (<5%) memory overhead over the standard training methods. The reason is perhaps that during training, the majority of the memory is for the dynamic computation of minibatch losses, whereas our method only adds to the static part of training. There is also no significant overhead for computation speed, perhaps for a similar reason. See the newly added experiment in Section B6, which shows that both the computation time overhead and memory overhead are less than 5% of that for vanilla training with a large batch size.

---

> > ### Author Response · Authors · 2024-11-26
> >
> > Hi! We noticed that you have not communicated with us during the discussion period. It would be really helpful to us if we could hear more from you. We would be happy to know any additional thoughts or feedback you have on our work and on our previous reply!

---

> > > ### Comment · Reviewer_Fj68 · 2024-11-27
> > > **Rebuttal Response**
> > >
> > > Thank you for the rebuttal and the response to my concerns. After going through the rebuttal, revised paper and all the other reviews, I have decided to maintain my score.

---

> > > > ### Comment · Area_Chair_Uzmf · 2024-11-27
> > > > **Reasons.**
> > > >
> > > > Dear Reviewer,
> > > > It would be helpful for the authors if you could reiterate any outstanding concerns or the primary reason for your score. Thanks!
> > > > -AC

---

> > > > > ### Author Response · Authors · 2024-12-01
> > > > > **Reply**
> > > > >
> > > > > Thank you AC for asking this question to reviewer Fj68.
> > > > >
> > > > > Reviewer Fj68: Even if you did not increase the score, we are interested hearing whether you think that our rebuttal has satisfactorily addressed your concerns. Also, if you have additional suggestions, we are happy to incorporate those into our final draft. Thanks!

---

### Author Response · Authors · 2024-11-22
**Summary rebuttal**

We thank all the reviewers for the detailed and constructive feedback. We are encouraged to hear that all reviewers find our manuscript positive, at least in some aspects. We would like to state our main contribution here, which some reviewers have misunderstood: we propose the first method that can remove almost all parameter symmetries without knowing anything about the symmetry. The method is theoretically guaranteed to work and is easy to implement (with essentially two-three lines of code).

To address the concerns of the referees, we have made the following changes and additions to the manuscript (colored in orange):

1. Removing symmetries in vision transformers in Section B6, which shows the methods can be easily scaled to the transformer architecture (Fj68, akeV)
2. Computation time/memory overhead experiment in Section B6, which shows that the proposed method has essentially no overhead in comparison to the standard training scheme when the batch size becomes large (Fj68, JSoA, akeV)
3. Empirical demonstration of the effects of Proposition 2, 3 for the science of deep learning in Section C (JSoA)
4. Additional test of the proposed method for the linear mode connectivity task, which shows competitive performance against the contemporaneous work of Lim et al. (2024), in section B7 (JSoA)



A major criticism comes from PYEZ. We believe the two points made by PYEZ are primarily due to a misunderstanding of our theoretical result and experimental settings:

1a. **The theory is too limited**. This is a misunderstanding. Our theory deals with the most general type of symmetry projections, not a restrictive subset of projections. The source of confusion is that we introduced a simplification to the definition of the symmetries. This simplification is justified in Proposition 1, which allows us to only focus on a subset of all symmetry projections without loss of generality. This is NOT a restriction of the theory.

1b. **Need to compare with weight decay**. This is also a misunderstanding. Every one of our experiments compares with weight decay (namely, the standard training method) with the same $\gamma$, which we refer to as the "vanilla" method (Figures 1, 2, 3, 5, 6, 7, 8, 9, 10, and 11). These comparisons are systematic and well-controlled in the sense that all other settings are equal. We have replaced "vanilla" with "wd" to avoid such a confusion.

---

> ### Author Response · Authors · 2024-11-22
> **Summary rebuttal part 2**
>
> Another major criticism comes from JSoA, which we believe are due to misunderstandings of our contribution and claims.
>
> 1a. **The claims are too strong**. JSoA argues that if we want to make strong claims about the performance, which we did not. Our primary claim is that we proposed the first algorithm that is capable of removing almost all symmetries in a neural network (or in any model), and this is justified in all of our experiments. We then showed that removing these symmetries correlates strongly with improved performances. Removing symmetries per se is an important and fundamental problem, a point acknowledged by reviewer akeV, for example.
> Our main claim is thus substantially justified by the experimental results and theory. At the same time, we never claimed that our method achieves SOTA performance in these tasks, and we do not think it is fair to criticize us for making strong claims.
>
> 1b. **Comparison with Lim et al. (2024) and other baselines for symmetry removal**. We believe that this is an unfair request for the following three reasons.
>
>
> (A) The comparison is inappropriate: it is essentially comparing apples to oranges. The method in Lim is a special-purpose method designed for removing permutation symmetry and for improving linear mode connectivity. In comparison, our method is designed to remove any generic symmetry, and its usage is far beyond the case of linear mode connectivity problems.
>
> (B) The comparison is unfair. Our method only has one hyperparameter to tune, whereas the method of Lim et al. requires two hyperparameters per layer (for W-asym). Even in the simplest five-layer MLP experiment, their method involves tuning 10 hyperparameters. A method with many hyperparameters cannot be compared fairly with a method with only one hyperparameter. Lim et al. also proposed a one-hyperparameter alternative ("$\sigma$-asym"), which we show below to not work as well as our method.
>
> (C) There is no other baseline. To the best of our knowledge, our method is the only one that removes almost every symmetry, and there is no other baseline that achieves this functionality. Rejecting our work based on comparisons that our method is not designed for is unreasonable. In fact, the only fair baseline is just the vanilla weight decay method, as our method can be seen as a simple variant of weight decay.
>
> (D) Even if we take a step back, the work of Lim et al. is a contemporaneous work to ours, and the ICLR rule does not require a comparison with Lim et al. The ICLR rule one comparison with previous work states that "We consider papers contemporaneous if they are published within the last four months. That means, since our full paper deadline is October 1, if a paper was published (i.e., at a peer-reviewed venue) on or after July 1, 2024, authors are not required to compare their own work to that paper." (https://iclr.cc/Conferences/2025/ReviewerGuide). The work of Lim et al. will be published in Neurips this year, and the neurips conference only starts in December (and the official publication date is after the conference). Even if we take the neurips author notification date as the publication date, it is still within a month of the ICLR submission deadline. Asking for this comparison is not fair, even according to the submission rules of ICLR.
>
> (E) In spite of all above, we compare \textit{syre} with Lim et al. (2024) for linear model connectivity and vision transformers. Our experiments suggest that \textit{syre} works similarly to Lim et al. (2024) for linear model connectivity with much fewer hyperparameters and works better ifor vision transformers.
>
> Lastly, we encourage and welcome the reviewers to raise additional questions, and we are happy to address them accordingly.

---

### Meta-Review · Area_Chair_Uzmf · 2024-12-19

**Metareview:**

**Summary** This paper studies the role of neural network parameter symmetries on the phenomenon of collapse in which the model is trapped in a low-capacity state during training.  The authors provide theory explaining how symmetry can lead to reduced capacity.  That propose a model-agnostic method called syre which is a variant of weight decay which can provably break symmetry during training and empirically leads to improved performance in domains where collapse has been an issue.

**Strengths** This paper provides an interesting and insightful theory linking parameter symmetries to reduced model capacity. This theory leads to a novel method, syre, which is simple, highly general, lightweight and can avoid symmetry and collapse. The empirical evaluations show syre is effective in practice in a range of settings (including for transformers after the rebuttal).

**Weaknesses** Reviewers initially had several criticisms, many of which were reduced by the author response.  Some reviewers considered the claims in the paper to be overstrong, but the authors reduced these claims in the revision.  Additionally, while syre is claimed to improve performance through symmetry reduction, the experiments in the paper hinged mainly on improved model performance, failing to demonstrate the exact mechanism.   The authors helped to address this by adding experiments showing syre can increase linear mode connectivity.

**Conclusion** The reviewers and the AC have a consensus that this paper provides an insightful theory connecting parameter symmetry to reduced model capacity and an effective method to mitigate the problem.  While there are potentially other experiments which can be performed with this method, I believe accepting the paper will help encourage that follow up work.

**Additional Comments On Reviewer Discussion:**

JSoA made several constructive critiques of the paper including clarifying the claims and adding additional tests on linear mode connectivity.  The authors made both such changes and JSoA increased their score.  PYEZ tested the authors claims against several examples.  The authors clarified their method and theory in the rebuttal and revision and PYEZ increased their score.

---

### Decision · Program_Chairs · 2025-01-22

Accept (Poster)